# Nonlinear reconciliation: Error reduction theorems

**Lorenzo Nespoli**                                                 *lorenzo.nespoli@supsi.ch*
*ISAAC*
*SUPSI*

**Anubhab Biswas**                                                 *anubhab.biswas@supsi.ch*
*ISAAC*
*SUPSI*

**Roberto Rocchetta**                                              *roberto.rocchetta@supsi.ch*
*ISAAC*
*SUPSI*

**Vasco Medici**                                                   *vasco.medici@supsi.ch*
*ISAAC*
*SUPSI*

**Reviewed on OpenReview:** *https://openreview.net/forum?id=dXRWuogm3J*

## Abstract

Forecast reconciliation, an ex-post technique applied to forecasts that must satisfy constraints, has been a prominent topic in the forecasting literature over the past two decades. Recently, several efforts have sought to extend reconciliation methods to the probabilistic settings. Nevertheless, formal theorems demonstrating error reduction in nonlinear constraints, analogous to those presented in Panagiotelis et al. (2021), are still lacking. This paper addresses that gap by establishing such theorems for various classes of nonlinear hypersurfaces and vector-valued functions. Specifically, we derive an exact analog of Theorem 3.1 from Panagiotelis et al. (2021) for hypersurfaces with constant-sign curvature. Additionally, we provide an error reduction theorem for the broader case of hypersurfaces with non-constant-sign curvature and for general manifolds with codimension $> 1$. To support reproducibility and practical adoption, we release a JAX-based Python package, JNLR, implementing the presented theorems and reconciliation procedures.

## 1 Introduction

Forecast reconciliation is a post-processing technique used to minimally correct an original set of forecasts, ensuring they satisfy known structural constraints. A straightforward example is the case of additive signals, grouped into increasingly larger sets, such as the quantity of a given good sold at the regional, state, and national levels. In such cases, the forecasts must obey additive relationships, a setting typically referred to as hierarchical forecasting (HF) in the literature.

Forecast reconciliation has become a widely adopted technique because it can be applied *ex-post*, independently of the forecasting model. This makes it a model-agnostic method for enforcing known structural constraints on forecasted signals. Reconciliation is typically motivated by two main considerations. An *existential* reason, if forecasts not respecting the known constraints are unusable or of little value. In this case reconciling is not an option. The second is a *utilitarian* reason: knowing the signals lie on a submanifold induces a bias in the production of the forecasts; this can lower the forecasting error. In the case of linear constraints, Theorem 3.1 of Panagiotelis et al. (2021) shows that orthogonally projecting forecasts onto the consistent subspace reduces the overall root mean squared error (RMSE) across the hierarchy. This refers to the RMSE of a single forecast vector at a given time point, computed as the square root of the

average of its squared components (as opposed to the RMSE over an entire test set). However, when the constraints are nonlinear, such a reduction is not always guaranteed: if reconciliation is performed solely to boost accuracy—rather than to strictly enforce the constraints—it can in fact lead to worse predictions. The ability to reliably estimate the probability of RMSE reduction under nonlinear constraints will significantly drive the adoption of reconciliation for improving forecasting accuracy. In this paper, we present theorems that establish conditions under which this RMSE reduction holds. In other words, these conditions answer the question: Should I expect an RMSE reduction when applying nonlinear reconciliation to the current forecasted point?

## 1.1 Linear reconciliation

Originally studied in industrial management as family-based forecasting (Fliedner, 2001), HF remains an active research topic, as evidenced by its inclusion in the 2020 M5 competition, where the organizers concluded that new methods capable of minimizing errors across all levels could offer significant accuracy gains (Makridakis et al., 2022). The simplest reconciliation strategy, bottom-up (Orcutt et al., 1968; Dunn et al., 1976), forecasts only the most disaggregated series, aggregating them upward. However, bottom-level series often have low signal-to-noise ratios, limiting performance. Forecasting all series independently leads to incoherent results unless reconciled. A major step forward was presented in (Hyndman et al., 2011), where the authors proposed an optimal reconciliation based on generalize least square (GLS).

Later refinements (Athanasopoulos et al., 2009; Wickramasuriya et al., 2019) proposed the use of covariance of base forecast errors. Later modification of the method include dropping the unbiasedness assumption (Ben Taieb et al., 2017; Di Modica et al., 2021), extension to temporal hierarchies, where series are aggregated over time (Athanasopoulos et al., 2017; Yang et al., 2017) and learning dynamic hierarchies (Cini et al., 2020; 2024). While point forecast reconciliation is well established, probabilistic reconciliation remained at first under-explored due to its complexity. Closed-form solutions exist only under strong assumptions (e.g., Gaussianity) (Corani et al., 2021), or errors' independence, when aggregation can be performed using convolutions. Other techniques based on sample reordering or copulas have been proposed (Jeon et al., 2019; Taieb et al., 2021), though they require knowledge of the joint cumulative density function (CDF) at each level or rely on empirical approximations. A recent framework reconciles samples from joint but incoherent distributions via constrained optimization based on scoring rules (Panagiotelis et al., 2023), outperforming copula methods but still assuming known joint CDF.

## 1.2 Non-linear reconciliation

The nonlinear reconciliation problem has been less studied in the context of forecasting. Since the nonlinear reconciliation problem doesn't usually have an analytical, closed-form solution, its application is less attractive than its linear counterpart. A method that could be used to produce a set of coherent forecasts for nonlinear manifolds, even if the authors applied it only to linear hierarchies, has been proposed by Spiliotis et al. (2021), and later applied in Rombouts et al. (2025), where a nonlinear regressor is trained to produce forecasts for the independent signal starting from the independent forecasts of all the signals. However, this method needs to train a machine learning model, and one has no control on the kind of projection applied, since this is implicitly learned by the model. On the other side, in the context of state estimation, nonlinear reconciliation techniques are well known. At the intersection of forecasting and nonlinear state estimation, the forecast aided state estimation (FASE) method has been proposed in the context of power grid state estimation (Brown Do Coutto Filho & De Souza, 2009). In FASE the forecasting model is a linear state-space, and the reconciliation step is equivalent to an iterated extended Kalman filter. In appendix A.1, we show that the FASE objective function represents a special case of the nonlinear reconciliation problem we introduce in the next section. That is, the theory presented in this paper is also valid for FASE and its application to power flow state estimation and forecasting.

## 1.3 Contributions

The main contributions are the following:

1. In section 2 we define the nonlinear reconciliation problem and an algorithm for its solution. In appendix A.1 we show that this algorithm is a generalization of the FASE algorithm.

2. In section 3 we present theorems providing point-wise RMSE reduction conditions for the the non-linear reconciliation problem, for different classes of manifolds $M$ implicitly defined by level sets of a function $f$. We propose two theorems, 1 and 2 for constant sign curvature manifolds and a third one, Theorem 3, providing a probabilistic reduction condition, which can be applied in the general case of $M$ with codimension $>1$ and without assumptions on sub-level sets of $f$.

3. In section 4 we empirically assessed their correctness. The results confirm the soundness of the theorems. Furthermore, we explore the possibility of using 3 to craft a reconciliation strategy beating the naive ones (always reconcile, never reconcile), and shows that these strategies lead to RMSE reduction.

4. To ease the application of these theorems and reconciliation methods, we release a JAX-based python library, JNLR https://github.com/supsi-dacd-isaac/JNLR.

## 2 Nonlinear reconciliation problem

We would like to generate predictions for $n$ signals, $\hat{z} \in \mathbb{R}^n$ which are known to satisfy one or more nonlinear constraints encoded by the level set $f(z) = 0$ of a vector-valued function $f : \mathbb{R}^n \to \mathbb{R}^m$ with $m$ continuous constraints $(1 < m \leq n)$. The nonlinear reconciliation problem minimally displacing the original forecasts $\hat{z}$ onto the constraint manifold $f(z) = 0$ can be written as:

$$\tilde{z} = \mathcal{S}(\hat{z}) := \arg\min_{z \in M} \|z - \hat{z}\|_W, \tag{1}$$

where we assume $M := \{z \in \mathbb{R}^n \mid f(z) = 0\}$ where $f : \mathbb{R}^n \to \mathbb{R}^m$ is smooth and 0 is a regular value of $f$. Under this assumption, $M$ is an $n - m$-dimensional differentiable manifold, with codimension $m$, embedded in $\mathbb{R}^n$. Here the dimension of $M$ refers to its intrinsic manifold dimension, which could differ from the ambient dimension in which it is embedded. We define the matrix norm as $\|z\|_W = z^T W z$ and $W \succ 0$ is a weight matrix assumed to be symmetric and positive definite (SPD). SPD assumption ensure a strictly positive objective. The mapping $\mathcal{S} : \mathbb{R}^n \to M$ is called the nonlinear reconciliation operator. The linear reconciliation problem is a subset of 1, where $M := \{z \in \mathbb{R}^n \mid z - A z_b = 0\}$, $A \in \mathbb{R}^{m \times (n-m)}$ and $z_b \in \mathbb{R}^{n-m}$ is usually referred to as bottom time series. The hierarchical reconciliation problem is a further subset, in which $A$ encodes a (usually unweighted) tree structure, $a_{i,j} \in \{0, 1\}$.

The nonlinear optimisation problem in equation 1 can be solved by forming the Lagrangian

$$\mathcal{L}(z, \lambda) = (z - \hat{z})^\top W (z - \hat{z}) + \lambda^\top f(z), \qquad \lambda \in \mathbb{R}^m. \tag{2}$$

A stationary point is characterised by $\nabla_z \mathcal{L} = 0$ and $\nabla_\lambda \mathcal{L} = 0$. Collecting these conditions gives the vector equation

$$F(z, \lambda) := \begin{bmatrix} \nabla_z \mathcal{L} \\ \nabla_\lambda \mathcal{L} \end{bmatrix} = \begin{bmatrix} 2W(z - \hat{z}) + J^\top \lambda \\ f(z) \end{bmatrix} = 0, \qquad J := \nabla f(z) \in \mathbb{R}^{m \times n}. \tag{3}$$

Starting from $(z_k, \lambda_k)$, a Newton–Raphson step $(\delta z, \delta \lambda)$ solves

$$\underbrace{\begin{bmatrix} 2W & J^\top \\ J & 0 \end{bmatrix}}_{J_F(z_k, \lambda_k)} \begin{bmatrix} \delta z \\ \delta \lambda \end{bmatrix} = -F(z_k, \lambda_k), \tag{4}$$

after which we update

$$(z_{k+1}, \lambda_{k+1}) = (z_k, \lambda_k) + (\delta z, \delta \lambda). \tag{5}$$

**Existence, uniqueness and convergence** Since $M = \{z \in \mathbb{R}^n \mid f(z) = 0\}$ is a closed set due to $f$ being continuous, and since the objective of equation 1 is coercive, a minimum for equation 1 always exists. Since the objective is convex, the solution is unique only if $M$ is convex. For the case of non-convex manifolds, under standard assumptions, namely i) $f \in C^2$, ii) the constraint Jacobian $J$ is full row rank: $\text{rank}(J(z_k)) = m$, iii) the reduced Hessian of the Lagrangian is positive definite on the tangent space at $\tilde{z}$, the Jacobian $J_F(\tilde{z}, \tilde{\lambda})$ is non-singular and the Newton iteration is locally quadratically convergent for initial guesses sufficiently close to $(\tilde{z}, \tilde{\lambda})$ where $(\tilde{z}, \tilde{\lambda})$ is a solution of Problem 1 (Nocedal & Wright, 2006; Dennis & Schnabel, 1983; Ortega & Rheinboldt, 1970).

In practice, for non-convex function, a globalization technique is usually used. In appendix A.5 we provide the pseudo-code of a solver using line search, which is also present in the released python package. This algorithm instantiates a classical augmented Lagrangian framework for equality-constrained nonlinear optimization, as described e.g. by and Bertsekas (2016) and Nocedal & Wright (2006). Under the same assumptions previously stated for local convergence of the Newton-Raphson solver and standard regularity assumptions of the augmented Lagrangian framework ((existence of a KKT point, Lipschitz continuity of the derivatives, and sufficiently accurate solution of the subproblems), augmented Lagrangian methods with line-search-based quasi-Newton inner solvers are known to be globally convergent in the sense that any accumulation point of the outer iterates satisfies the KKT conditions of the original constrained problem. We do not derive new convergence results here, but we rely on this well-established theory; in our experiments, the algorithm converged robustly for all tested forecasts.

## 3 Error reduction theorems

In the case of linear constraints, it has been shown that reconciliation via orthogonal projection always reduces the overall point-wise prediction error. This result relies on the Pythagorean theorem and was demonstrated in Panagiotelis et al. (2021). In this section, we derive analogous theorems for nonlinear constraints.

All proofs are based on a geometric interpretation of problem 1, as illustrated in Figure 1, which depicts the reconciliation process for two examples of predicted points, denoted by $\hat{z}$.

Let $\hat{z}$ be a prediction of the true point $z$, and let $\tilde{z}$ denote the orthogonal projection of $\hat{z}$ onto the manifold $M$. We define three key vectors: the original prediction displacement $\hat{\delta} = z - \hat{z}$, the reconciled displacement after projection $\tilde{\delta} = z - \tilde{z}$, and the reconciliation adjustment $\delta_\pi = \tilde{z} - \hat{z}$.

The condition for RMSE reduction can then be stated as:

$$\|\hat{\delta}\|^2 \geq \|\tilde{\delta}\|^2 \tag{6}$$

Using the vectorial relation $\hat{\delta} = \delta_\pi + \tilde{\delta}$ this becomes:

$$\|\delta_\pi\|^2 + 2\delta_\pi^T \tilde{\delta} \geq 0 \tag{7}$$

that can be rewritten as:

$$\delta_\pi^T \tilde{\delta} \geq -\frac{\|\delta_\pi\|^2}{2} \tag{8}$$

### 3.1 Constant sign curvature hypersurfaces

A hypersurface in $\mathbb{R}^n$ is a smooth submanifold with codimension 1 that can be locally defined as the level set $M = \{z \in \mathbb{R}^n : f(z) = c\}$, where $f : \mathbb{R}^n \to \mathbb{R}$ is a smooth function and $c$ is a regular value (i.e., $\nabla f(z) \neq 0$ for all $z \in M$). As an example, Figure 1 shows the hypersurface given by $M = \{z \in \mathbb{R}^2 : f(z) \triangleq y - x^2 = 0\}$ as a thick black line, while blue and violet represent regions of the negative and positive values of $f(z)$.

For hypersurfaces with a constant sign curvature, we can exploit the fact that they possess a supporting hyperplane at the point of projection. If the forecasted point $\hat{z}$ is on the right side of the hypersurface, it will see the surface *bending away* from it, which is sufficient to guarantee a reduction of the error *independently* from where the true point $z$ is located.

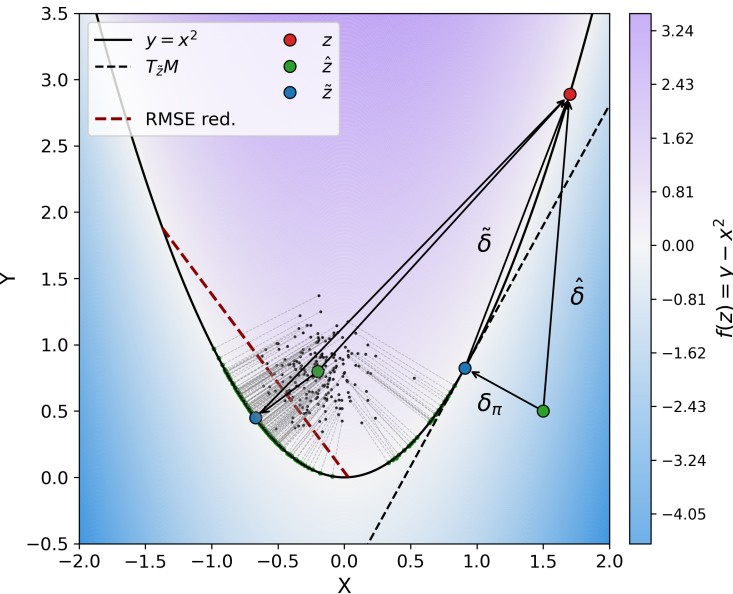

Figure 1: Conceptual plot for Theorem 1 ($\hat{z}$ with negative $f(\hat{z})$ in the bottom right) and Theorem 3 ($\hat{z}$ with positive $f(\hat{z})$, shown along atoms of its predictive distribution and their projection onto $M$).

---

**Theorem 1: Constant sign curvature hypersurfaces**

*Let $M = \{z : f(z) = 0\} \subset \mathbb{R}^n$ be a hypersurface defined by $f : \mathbb{R}^n \to \mathbb{R} \in C^2$ with $\nabla f(z) \neq 0$ on $M$ and with convex sub or super level sets. Given a prediction $\hat{z} \in \mathbb{R}^n$, denote by $\tilde{z} \in M$ its orthogonal projection onto $M$. A sufficient local condition, that can be computed from the value of $f$ at the forecasted point $\hat{z}$ and the Hessian at the projection point $\tilde{z}$, for the reduction of the RMSE after projecting $\hat{z}$ into $\tilde{z}$ is the following:*

$$\text{sign}(f(\hat{z})) \cdot \lambda_{min}(H_{tan}(\tilde{z})) > 0 \tag{9}$$

*where $H_{tan}$ denotes the Hessian $H = D^2 f(z)$ restricted to the tangent space of $M$ (see appendix A.2) and $\lambda_{min}$ denotes its smallest eigenvalue.*

---

**Proof 1** *A bounding condition for equation 7 to be positive is $\delta_\pi^T \tilde{\delta} \geq 0 \ \forall \ \tilde{\delta}$. That is, if we can show that this condition holds for arbitrary values of $\tilde{\delta}$ the theorem holds. We proceed in this way since $\tilde{\delta}$ depends on where the true point will be on the manifold, which is unknown at prediction time, so the need of bounding the expression for any value of $\tilde{\delta}$. Since $\delta_\pi$ is an orthogonal projection onto $M$, it is parallel to the gradient at $\tilde{z}$, so we can rewrite*

$$\mu \nabla f(\tilde{z})^T \tilde{\delta} \geq 0 \quad \forall \tilde{\delta} \tag{10}$$

*The sign of $\mu$ is given by the side $\hat{z}$ is located with respect to the level set $f(z) = 0$, that is*

$$sign(\mu) = -sign(f(\hat{z})) \tag{11}$$

*since if $f(\hat{z})$ is positive, the projecting vector $\delta_\pi$ will oppose the gradient and vice versa.*

*Since $M$ is the boundary of a convex sub or super level set of $f(z)$, it has a supporting hyperplane in $\tilde{z}$, that implies the sign of $\nabla f(\tilde{z})^T \tilde{\delta}$ is constant on $M$ w.r.t. $\tilde{\delta}$. This also implies:*

$$sign(\nabla f(\tilde{z})^T \tilde{\delta}) = -sign(\lambda_{min}(H_{tan}(\tilde{z}))) \tag{12}$$

*(see A.3 for the proof). Putting together equation 10 equation 11 and equation 12 we obtain the sufficient condition of Theorem 1.*

> **Corollary 1: Constant sign curvature hypersurfaces**
>
> *Under the same assumptions of Theorem 1, defining $\delta_\pi = \mu \nabla f(\tilde{z})$, orthogonally projecting $\hat{z}$ into $\tilde{z}$ will decrease RMSE if the following condition holds:*
>
> $$\mu = \begin{cases} < 0 & if \quad \{z : f(z) < 0\} \ convex \\ > 0 & if \quad \{z : f(z) > 0\} \ convex \end{cases} \tag{13}$$

**Proof of corollary 1**   *The projection vector $\delta_\pi$ at $\tilde{z}$ is by definition parallel to $\nabla f(\tilde{z})$ so that $\delta_\pi = \mu \nabla f(\tilde{z})$ holds. Since sub-level set of $f(z)$ is convex iff $\lambda_{min}(H_{tan}) > 0$ (see A.2), considering equation 11 we can re-write condition 9 as equation 13.*

Corollary 1 states that, knowing if the sub or super level sets of $f(z)$ are convex, we can restate Theorem 1 using the gradient at $\tilde{z}$ without computing the Hessian. We will use this fact to craft a theorem for vector valued functions in the next section.

### 3.2   Multiple constraints

The next result extends Corollary 1 to manifolds defined by multiple constraints, or vector-valued functions. The key idea is the same: if each defining function $f_i$ has convex sub or super level set, and if the projection direction lies in the positive cone spanned by the gradients at projection point $\{\nabla f_i(\tilde{z})\}_{i=1}^m$, then this direction defines a supporting hyperplane to the intersection $C = \cap_{i=1}^m C_i$ of the corresponding convex sets $C_i = \{z : f_i(z) \le 0\}$. This yields a uniform decrease of squared error. Figure 2 illustrates the construction.

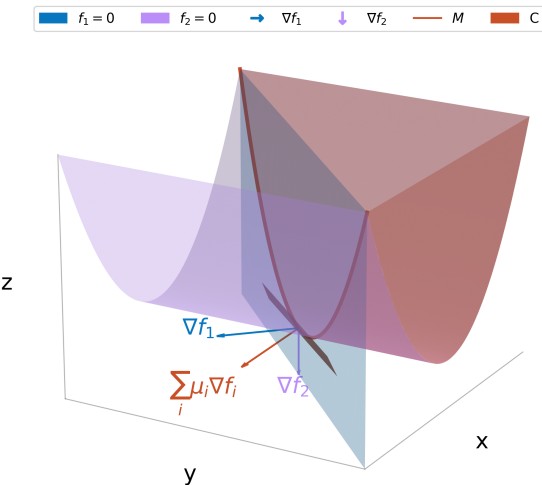

Figure 2: Concept for Theorem 2 with $f = (f_1, f_2) = [x - y, \ x^2 - z]$. The zero level sets $f_1 = 0$ and $f_2 = 0$ are shown in blue and violet. The intersection $C = \{f_1 \le 0\} \cap \{f_2 \le 0\}$ is in green. If the projection direction is a positive combination of the normals $\nabla f_1(\tilde{z}), \nabla f_2(\tilde{z})$, it defines a supporting hyperplane for $C$ (gray).

> **Theorem 2: Manifolds of codimension $m > 1$ with convex/concave level geometry**
>
> *Let $M = \{z \in \mathbb{R}^n : f(z) = 0\}$ with $f = (f_1, \ldots, f_m) : \mathbb{R}^n \to \mathbb{R}^m$, $f \in C^2$, and assume $\nabla f_i(z) \ne 0$ and $\operatorname{rank} J(z) = m$ on $M$, where $J(z) = \nabla f(z)$. Suppose that for each $i$, the sublevel set $C_i = \{z : f_i(z) \le 0\}$ is convex. For any $\hat{z} \in \mathbb{R}^n$ let $\tilde{z} \in M$ be its orthogonal projection and define*
>
> $$\delta_\pi := \tilde{z} - \hat{z}, \qquad \tilde{\delta}(z) := z - \tilde{z}.$$

> *If there exists $\mu \in \mathbb{R}_+^m$ such that*
>
> $$\delta_\pi = -\sum_{i=1}^m \mu_i \nabla f_i(\tilde{z}), \tag{14}$$
>
> *then for every $z \in C := \cap_{i=1}^m C_i$, $\|z - \hat{z}\|^2 \geq \|z - \tilde{z}\|^2$; in particular, the inequality holds for all $z \in M$.*
>
> *If instead each $f_i$ is concave (equivalently, the superlevel sets $\{f_i \geq 0\}$ are convex), the same conclusion holds with the sign in equation 14 reversed: $\delta_\pi = \sum_{i=1}^m \mu_i \nabla f_i(\tilde{z})$.*

**Proof 2** *Fix $\tilde{z}$ and let $z \in C := \cap_{i=1}^m C_i$. Since $C_i$ is convex and $f_i(\tilde{z}) = 0$, the hyperplane with normal $\nabla f_i(\tilde{z})$ supports $C_i$ at $\tilde{z}$:*

$$\nabla f_i(\tilde{z})^\top (z - \tilde{z}) \leq 0 \quad \text{for all } z \in C_i,$$

*hence also for all $z \in C$. If equation 14 holds with $\mu_i \geq 0$, then*

$$\delta_\pi^\top \tilde{\delta}(z) = -\sum_{i=1}^m \mu_i \nabla f_i(\tilde{z})^\top (z - \tilde{z}) \geq 0.$$

*Using $\hat{\delta} = \delta_\pi + \tilde{\delta}(z)$,*

$$\|\hat{\delta}\|^2 - \|\tilde{\delta}(z)\|^2 = \|\delta_\pi\|^2 + 2\delta_\pi^\top \tilde{\delta}(z) \geq \|\delta_\pi\|^2 \geq 0,$$

*which proves the claim. In the concave case, the supporting inequality reverses sign for the convex superlevel sets, yielding the stated flip of equation 14.*

## 3.3 Applicability of Theorem 1 and 2

Theorems 1 and 2 provide *sufficient* conditions for a decrease of RMSE, conditioned on whether the forecast $\hat{z}$ lies in the relevant sublevel or superlevel region of the function(s) defining $M$. How often those conditions hold depends on the geometry of $M$.

We estimate an *a priori* likelihood as follows. For several hypersurfaces ($m = 1$) and manifolds of codimension $m > 1$, we sample points $z_i \in M$. For each $z_i$ we draw an isotropic error $\varepsilon_i \sim \mathcal{N}(0, \Sigma)$ with $\Sigma = \sigma_I^2 I_n$ at increasing noise standard deviations $\sigma_I$, form the forecast $\hat{z}_i = z_i + \varepsilon_i$, and compute its projection $\tilde{z}_i$ by solving equation 1. We then: (i) check the corresponding theorem's condition at $\tilde{z}_i$ and record the indicator of it holding, and (ii) record whether reconciliation reduces squared error, i.e. $\mathbf{1}\{\|z_i - \tilde{z}_i\|^2 < \|z_i - \hat{z}_i\|^2\}$. Averaging these indicators over $i$ yields the empirical probabilities shown in Figure 3 (left: theorem's condition holds; right: squared-error reduction).

Across the tested manifolds, the sufficient conditions identify only a subset of cases that actually benefit from reconciliation. Moreover, real forecasting errors are rarely isotropic around a nonlinear $M$. The next example shows that even with optimal mean–squared prediction, forecasts can systematically fall into the region where Theorems 1 and 2 do *not* trigger.

> **Example 1: Optimal prediction on a parabola**
>
> *Let $M = \{z : f(z) = 0\}$ be a parabola, with $f(z) = z_1^2 - z_2 : \mathbb{R}^2 \to \mathbb{R}$ where $z = (z_1, z_2) \in \mathbb{R}^2$. Given a dataset $\mathcal{D} = \{(z_{1,i}, z_{2,i})\}_{i=1}^n$, the independent least-squares predictors are the empirical means*
>
> $$\hat{z}_1^* = \tfrac{1}{n}\sum_i z_{1,i}, \qquad \hat{z}_2^* = \tfrac{1}{n}\sum_i z_{2,i} = \tfrac{1}{n}\sum_i z_{1,i}^2 \geq \left(\tfrac{1}{n}\sum_i z_{1,i}\right)^2 = \hat{z}_1^{*2},$$
>
> *where the inequality is Jensen's. Hence $f(\hat{z}^*) = \hat{z}_1^{*2} - \hat{z}_2^* \leq 0$, i.e., the optimal independent prediction lies in the sublevel region $\{z_2 \geq z_1^2\}$.*

Therefore, the rates in Figure 3 are optimistic regarding the frequency with which the sufficient conditions will hold in practice: in many realistic settings, $\hat{z}$ is likely to lie in the region where a reduction in error

is not guaranteed by Theorems 1 and 2. This motivates the probabilistic guarantees for general nonconvex manifolds of codimension $m > 1$ developed in the next section.

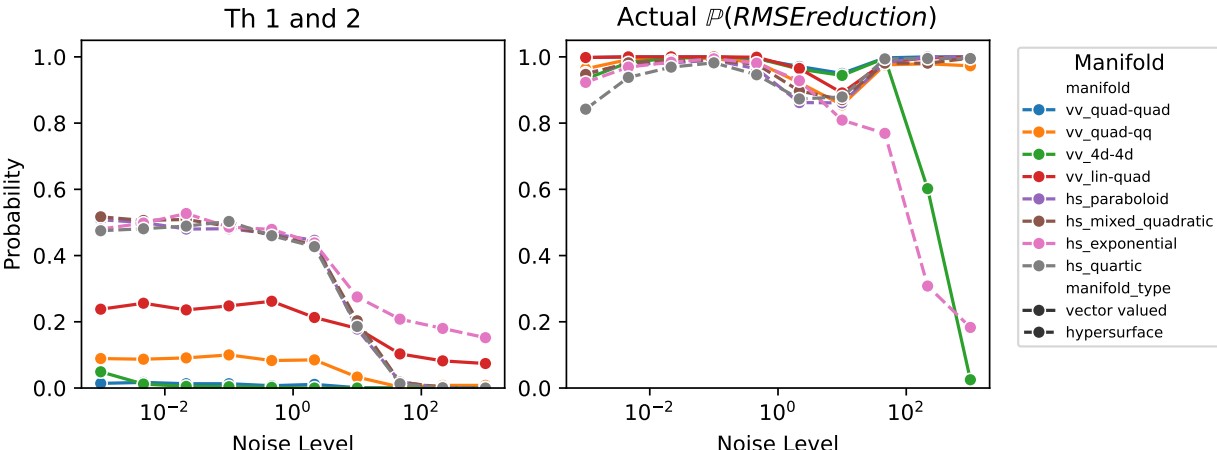

Figure 3: Left: probability of observing a positive condition from Theorem 1 and 2 under isotropic Gaussian perturbation of points on the manifold. Right: empirical probability of reducing RMSE by reconciling. The x axis shows the noise level $\sigma_I$.

## 3.4 Non-constant sign curvature manifolds and probabilistic estimators

In Theorem 1 we used the orientation condition in equation 11 together with equation 12 to obtain a sufficient error–reduction result for constant-sign curvature hypersurfaces. However, as shown in figure 1, these conditions are conservative: if the prediction $\hat{z}$ lies on the side where $M$ bends *towards* $\hat{z}$, the Theorem 1 can't predict if the RMSE is going to decrease. For those cases, and for general manifolds without constant-sign curvature, strictly deterministic guarantees are not possible without knowing the true location of $z$.

However, if we shift the perspective from deterministic guarantees to *probabilistic* ones, we can still leverage the general inequality described in equation 8, which says that the post-projection error vector $\tilde{\delta}$ may "climb" along the projection direction $\delta_\pi$ by at most half its length. Geometrically, this defines a half-space of candidate true points $z$ for which reconciliation reduces the RMSE. Figure 1 illustrates, in two dimensions, the corresponding half-plane for the left prediction point. For surfaces that can be sampled, this half-space intersects $M$ in a region where we will observe a reduction.

While informative, the geometric condition in equation 8 does not directly quantify the likelihood of error reduction. We propose a practical probabilistic estimator to answer: *What is the probability that projecting the prediction onto $M$ reduces RMSE?.* Since the only unknown in equation 8 is the post-projection error $\tilde{\delta}$, which is fully determined by $\tilde{z}$ and $z$, this question reduces to estimating the probability that the true point lies in the favorable half-space. The following discussion is general in terms of reconciliation method; in particular *we don't assume an orthogonal projection for the reconciliation.*

### 3.4.1 Coherent probability distributions on nonlinear manifolds

We recall the definition of reconciled probability distribution introduced in Panagiotelis et al. (2023) and Zambon et al. (2023). This concept underpins the probabilistic guarantees we derive for manifolds with non-constant-sign curvature.

> **Definition 1: Coherent probability distribution on a nonlinear manifold**
>
> Let $\hat{\nu} \in \mathcal{P}(\mathbb{R}^n)$ be a predictive distribution on $(\mathbb{R}^n, \mathcal{B}_{\mathbb{R}^n})$ and let $\mathcal{S} : \mathbb{R}^n \to M$ be a Borel-measurable reconciliation operator whose image equals the constraint manifold $M := \{z \in \mathbb{R}^n : f(z) = 0\}$. The coherent predictive distribution $\tilde{\nu} \in \mathcal{P}(M)$ is the pushforward
>
> $$\tilde{\nu} = \mathcal{S}_{\#}\hat{\nu}, \qquad \tilde{\nu}(F) = \hat{\nu}\big(\mathcal{S}^{-1}(F)\big), \quad F \in \mathcal{B}_M,$$
>
> where $\mathcal{B}_M := \{F \subseteq M : \mathcal{S}^{-1}(F) \in \mathcal{B}_{\mathbb{R}^n}\}$.

### 3.4.2 Probability of RMSE reduction

Let $h : \mathbb{R} \to \{0, 1\}$ be the Heaviside function $h(u) = \mathbf{1}_{\{u>0\}}$. For each time $t$, let $Z_t$ be a random vector with (unknown) true and predictive distribution $\nu_t, \tilde{\nu}_t$ on the ambient space, and let $\tilde{z}_t$ denote the (deterministic) reconciled point at time $t$. We assume $\nu_t, \tilde{\nu}_t$ are absolutely continuous w.r.t. a reference measure $\lambda$ such that their probability density functions exist, $\frac{d\nu_t}{d\lambda} := f_{\nu_t}$ $\frac{d\tilde{\nu}_t}{d\lambda} := f_{\tilde{\nu}_t}$. We write $\delta_{\pi,t} := \tilde{z}_t - \hat{z}_t$ for the reconciliation adjustment. Define the scalar map:

$$\phi_t(z) := \delta_{\pi,t}^\top(z - \tilde{z}_t) + \frac{1}{2}\|\delta_{\pi,t}\|^2 \tag{15}$$

From condition equation 8, projection from $\hat{z}_t$ to $\tilde{z}_t$ reduces the RMSE at time $t$ if and only if $\phi_t(Z_t) > 0$. Thus the *true probability of RMSE reduction* at time $t$ is:

$$e_t := \nu_t\big(\Phi_t > 0\big) = \int_M h(\phi_t(z)) \, d\nu_t(z) = \int_{\mathbb{R}} h(y) \, d\mu_t(y) = \mathbb{E}_{\mu_t}[Y_t], \tag{16}$$

where $\mu_t = \phi_{t,\#}\nu_t$ is the distribution of the scalar random variable $\Phi_t := \phi_t(Z_t)$ on $\mathbb{R}$, and $Y_t := h(\Phi_t)$ is the binary indicator of reduction.

In practice, the true law $\nu_t$ is never observed, and we just have access ex-post to a sample drawn from it at time t, $z_t \sim \nu_t$. We therefore approximate $\nu_t$ by a consistent *predictive distribution* $\tilde{\nu}_t$, from which we can retrieve an estimated probability of improvement $\tilde{e}_t := \mathbb{E}_{\tilde{\nu}_t}[h(\phi_t(Z_t))]$. To ease geometric intuition, Figure 4 depicts the relationship between $\nu_t, \tilde{\nu}_t, \mu_t, \tilde{\mu}_t, \phi_t$ and the estimators $e_t, \tilde{e}_t$.

### 3.4.3 Monte Carlo Estimator

Given $S$ samples (atoms) $\tilde{z}_{t,s} \sim \tilde{\nu}_t$ with normalized weights $\tilde{\pi}_{t,s}$ (where $\tilde{\pi}_{t,s} = 1/S$ for unweighted Monte Carlo) we define the estimator:

$$\tilde{e}_{mc,t} := \frac{1}{S}\sum_{s=1}^{S} h\big(\phi_t(\tilde{z}_{t,s})\big). \tag{17}$$

Without loss of generality, and to ease the notation, we will just consider the unweighted Monte Carlo.

> **Lemma 1: Consistency of the Estimator**
>
> Let $\tilde{\nu}_t$ be a predictive distribution at time $t$, and let $\tilde{z}_{t,1}, \ldots, \tilde{z}_{t,S}$ be i.i.d. draws from $\tilde{\nu}_t$. Then $\tilde{e}_{mc,t}$ (with equal weights) is an unbiased and strongly consistent estimator of $\tilde{e}_t$ under the predictive law $\tilde{\nu}_t$, i.e., $\mathbb{E}[\tilde{e}_{mc,t}] = \tilde{e}_t$ and $\tilde{e}_{mc,t} \xrightarrow{a.s.} \tilde{e}_t$ as $S \to \infty$.

By construction, $Y_s$ are i.i.d. with $\mathbb{E}[Y_s] = \tilde{e}_t$ thus the result follows immediately from the Strong Law of Large Numbers.

One possible practical procedure to obtain $\tilde{e}_{mc,t}$ is the following:

1. Obtain the reconciled point forecast via the nonlinear projection operator: $\tilde{z}_t = \mathcal{S}(\hat{z}_t)$

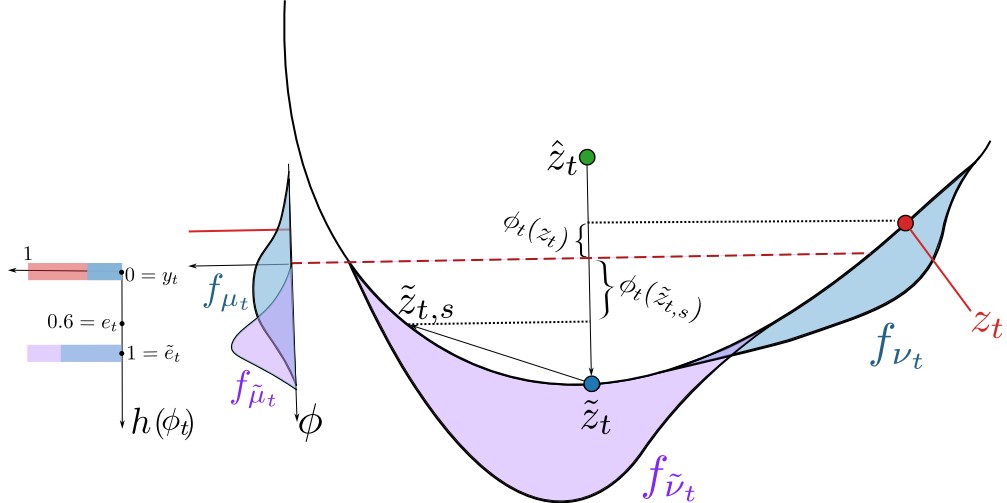

Figure 4: Conceptual diagram for the probabilistic RMSE reduction. The unknown true distribution $\nu_t$ and the predictive distribution $\tilde{\nu}_t$ are mapped by $\phi_t$ to the scalar probability density functions of $\mu_t := \phi_{t,\#}\nu_t$ and $\tilde{\mu}_t := \phi_{t,\#}\tilde{\nu}_t$. Applying $h$ yields the true and predicted probabilities of RMSE reduction $e_t = \mathbb{E}_{f_{\mu_t}}[h(\Phi_t)]$ and $\tilde{e}_t = \mathbb{E}_{f_{\tilde{\mu}_t}}[h(\tilde{\Phi}_t)]$, respectively. In the depicted realization, 60% of $f_{\mu_t}$ is below the dashed line defined by $\phi_t(z_t) = 0$, so the true RMSE reduction probability is $e_t = 0.6$. However, the predictive distribution entirely lies below the dashed line, resulting in an estimated probability $\tilde{e}_t = 1$. The observed $z_t$, lies above the dashed line, so $\phi_t(z_t) < 0$ and the projection to $\tilde{z}_t$ would *increase* the RMSE at time $t$ (and thus $y_t = 0$).

2. Starting from atoms of a non-reconciled predictive distribution $\hat{z}_{t,s}$, obtain the consistent samples always via projection $\tilde{z}_{t,s} = \{\mathcal{S}(\hat{z}_{t,s})\}_{s=1}^{S}$. By definition 1 this defines a sample-based representation of a coherent probability distribution on $M$.

3. Count how many reconciled samples fall in the right half-space via equation 17, using $\tilde{\delta}_{t,s} = \tilde{z}_{t,s} - \tilde{z}_t$

We emphasize that the estimator equation 17 does not require the use of an orthogonal projection, in contrast to Theorem 1 and Theorem 2. This represents a distinct advantage, since, at least in the case of linear constraints, non-orthogonal projections have been shown to yield even greater RMSE reductions in many settings.

### 3.4.4 Calibration Bounds

While Lemma 1 guarantees convergence to the *predicted* probability $\tilde{e}_t$, it does not guarantee that $\tilde{e}_t$ matches the *true* probability $e_t$. A more interesting result would be to bound the distance of the Monte Carlo estimator $\tilde{e}_{mc,t}$ from the realized outcome $Y_t$, $|\tilde{e}_{mc,t} - Y_t|$, or the distance of the Monte Carlo estimator from the true probability of RMSE reduction, $|\tilde{e}_{mc,t} - e_t|$. The distance $|\tilde{e}_{mc,t} - Y_t|$ cannot be bounded tightly for a single instance $t$ because we only observe a single binary outcome $Y_t = h(\phi_t(Z_t))$ and never have access to the true distribution $\nu_t$ (see appendix A.4). However, we can derive rigorous bounds on the *calibration* of the estimator $\tilde{e}_{mc,t}$ over an archive of forecasts.

---

**Theorem 3: Finite-Sample Conditional Calibration Bounds**

*Let us define the **true conditional probability of RMSE reduction** at prediction level $e$ as:*

$$\pi(e) := \mathbb{P}(Y = 1 \mid E = e), \tag{18}$$

*For any $e \in (0,1)$, we define the binned archive*

$$\mathcal{A}_{\Delta}^{N}(e) := \big\{(e_i, y_i) : i \in \{1, \dots, N\}, \ e_i \in [e - \Delta, e + \Delta]\big\}. \tag{19}$$

---

*with bin size $N_\Delta(e) := |\mathcal{A}_\Delta(e)|$ and number of successes $k_\Delta(e) := \sum_{i \in \mathcal{A}_\Delta(e)} Y_i$, with $y_i \in \{0,1\}$ a binary indicator for the successful reconciliation and $e_i$ error reduction probability under the forecast law.*

**I - Bounds for $\pi(e)$:** *Assume each bin $A_\Delta^N(e)$ contains i.i.d. scenario pairs. Then, with probability at least $1-\beta$ over the random mechanism producing $A_\Delta^N(e)$, the true conditional probability is bounded by*

$$e_L(\beta, N_\Delta(e), k_\Delta(e)) \leq \pi(e) \leq e_U(\beta, N_\Delta(e), k_\Delta(e)). \tag{20}$$

*where $[e_L, e_U]$ denotes the Clopper–Pearson confidence interval for a binomial proportion with $N_\Delta(e)$ trials and $k_\Delta(e)$ observed successes at confidence level $1 - \beta$.*

**II - Instantaneous bounds for $|e_t - \tilde{e}_{mc,t}|$:** *Assume the conditional distribution is stationary such that $e_t = \pi(e)$ Consequently, for a deployment time $t$ where equation 17 predicts $\tilde{e}_{mc,t} = e$, the instantaneous estimation error is bounded by:*

$$L_{\text{err}} \leq |e_t - \tilde{e}_{mc,t}| \leq U_{\text{err}}, \tag{21}$$

*where the magnitude bounds are determined by the geometry of the interval relative to the estimate $e$:*

$$U_{\text{err}} := \max(|e_U - e|, |e_L - e|), \tag{22}$$

$$L_{\text{err}} := \begin{cases} e_L - e & \text{if } e_L > e, \\ e - e_U & \text{if } e_U < e, \\ 0 & \text{otherwise (if } e \in [e_L, e_U]). \end{cases} \tag{23}$$

**Proof 3** *$\mathbf{I}$ - **Bounds for** $\pi(e)$ We condition on the random event $E = e_i$ for all $e_i \in \mathcal{A}_\Delta^N(e)$ and, under the i.i.d. assumption of the archive, the outcomes $y_i$ are independent Bernoulli trials with success probability $\pi(e)$ defined in equation 18. Therefore, the count of successes follows a Binomial distribution: $K \sim \text{Bin}(k, \pi(e))$. The Clopper–Pearson interval $[e_L, e_U]$ is constructed by inverting the Binomial hypothesis test. By definition (Clopper & Pearson, 1934), it satisfies:*

$$\mathbb{P}_{K \sim \text{Bin}(k, \pi(e))}\Big(\pi(e) \in [e_L(K), e_U(K)]\Big) \geq 1 - \delta.$$

*This establishes that the random interval $[e_L, e_U]$ covers the true conditional probability $\pi(e)$ with probability at least $1 - \delta$.*

**II - Instantaneous bounds for $|e_t - \tilde{e}_{mc,t}|$** *Under the stationarity assumption, the true probability at time $t$ is $e_t = \pi(e)$. We condition on the event that the interval covers the truth, i.e., $e_L \leq e_t \leq e_U$, which holds with probability $\geq 1 - \delta$. The estimation error is $|e_t - \tilde{e}_{mc,t}|$. Since $\tilde{e}_{mc,t} = e$ is fixed and $e_t$ is constrained to the interval $[e_L, e_U]$, the error is constrained by the minimum and maximum distances from the scalar $e$ to the set $[e_L, e_U]$.*

- *The maximum distance $U_{\text{err}}$ occurs at one of the endpoints, hence $U_{\text{err}} = \max_{p \in [e_L, e_U]} |p - e| = \max(|e_L - e|, |e_U - e|)$.*

- *The minimum distance $L_{\text{err}}$ is 0 if $e$ lies inside the interval. If the interval is entirely above $e$ ($e_L > e$), the minimum distance is $e_L - e$. If the interval is entirely below $e$ ($e_U < e$), the minimum distance is $e - e_U$.*

*Combining these cases yields the definitions in equation 22 and equation 23.*

**Use and data collection:** Informative reconciliation guarantees arise when the interval $[e_L, e_U]$ is narrow and statistical uncertainty is low. For example, a one-sided lower bound $\mathbb{P}[\pi(e) \geq e_L]$ with $e_L \geq 0.9$ provides

strong evidence in favor of reconciliation. Similarly, an upper bound $\mathbb{P}[\pi(e) \leq e_U]$ with $e_U \leq 0.1$ indicates that reconciliation is unlikely to be beneficial. Both cases are informative from a statistical standpoint. In contrast, an interval $[e_L, e_U] = [0.1, 0.9]$ offers a weak statistical support and does not convey a clear expectation of positive error reduction. On the other hand, if the interval is narrow but close to 0.5, e.g., $[e_L, e_U] = [0.45, 0.55]$, the statistical (epistemic) uncertainty in the estimate is low. Still, the results of the previous theorems do not provide a definitive indication of whether reconciliation should be applied in this case. This is due to the geometry of the problem (manifold) and natural variability in the probabilistic predictions and data. Aiming for a reliable reconciliation with a high confidence, that is, high $e_L$ and low $\beta$, an archive $\mathcal{A}_\Delta^{N^\star}(e)$ with minimum size $N^\star$ is required. The data set size may be computed numerically by inverting Theorem 3; however, this requires specifying a target $k$ (or expected rate $k/N$), which is not always controllable. Yet inversion offers a simple and practical guideline for experimental design and selecting a sufficiently large sample size for a target $e$. In fact, the minimal required sample size $N^\star$ may be increased by further exploring the data-generating process and enlarging the archive, i.e., by collecting additional scenario pairs $(e_i, y_i)$ to verify whether reconciliation effectively reduces the error. This approach can help mitigate epistemic uncertainty by providing a more representative coverage of the forecast law at the target prediction level $e$. However, a practical limitation arises when the forecaster, data, reconciliation, and manifold jointly lead to a limited number of hits within a specific bin $e_i \in [e - \Delta, e + \Delta]$. In such cases, even increasing the overall archive may not sufficiently decrease uncertainty for that bin, as the number of realizations directly constraining $\pi(e)$ is inherently limited. Consequently, while enlarging $N^\star$ is a viable strategy to strengthen statistical guarantees, the achievable reduction in epistemic uncertainty is ultimately constrained by the intrinsic distribution of forecasted errors and the underlying geometry of the forecaster and the manifold. This underscores the importance of both careful experimental design and awareness of the process-specific limitations when interpreting informative reconciliation guarantees.

## 4    Numerical tests

To facilitate the application of the proposed theorems, we provide an open-source JAX-based Python library, https://github.com/supsi-dacd-isaac/JNLR, which supports just-in-time compilation and GPU acceleration. We use this library to conduct numerical tests for the presented theorems. The tests cover a collection of manifolds with codimension 1 and 2, including cases with constant-sign curvature and convex sub-level sets. The applicability of each theorem to the different classes of manifolds is summarized in Table 1.

Table 1: Applicability of the presented theorems.

|  | constant sign curvature | non-constant sign curvature |
| --- | :---: | :---: |
| codimension 1 | 1 2 3 | 3 |
| codimension >1 | 23 | 3 |

Furthermore, as stated before, equation 17 also applies when using non-orthogonal projections.

### 4.1    Experiments setup and evaluation metrics

The goal of the numerical tests is to evaluate the presented theorems in a realistic forecasting scenario. Reconciliation is usually used as a post-processing technique, after each of the $n$ variables of the multidimensional point $z$ has been predicted independently by a statistical or machine-learning model. Since predictions are independently generated, they won't perfectly lie on the manifold $M$ defined by the constraints. Reconciliation corrects this by projecting an unconstrained forecast onto $M$, enforcing coherence. The experiments in this section simulate a typical forecasting pipeline. We start by generating data from a autoregressive process. This allows us to control the variance and correlation of the different components of $z$ and change them systematically. This is a standard setting in simulation studies for forecast applications. Secondly, we independently fit $n$ standard regression models (LightGBM), which are then used to obtain our unreconciled

forecasts $\hat{z}$. Those are then projected via $\mathcal{S}$, which allows us to investigate the validity of the RMSE reduction theorems. A more technical description of the setup follows.

All the tested manifolds are defined by one or two graphs $g : \mathbb{R}^2 \to \mathbb{R}$, so that $M : \{z \in \mathbb{R}^2 : f(z) = 0\}$, $f = g(z_1, z_2)$ for manifolds with codimension 1 and $f = (g_1(z_1, z_2), g_2(z_1, z_2))$ for manifolds with codimension 2. This allows us to sample the manifolds by defining a two-dimensional data generation process spanning the surfaces:

$$z_1^{t+1} = \theta_1 z_1^t + w_1$$
$$z_2^{t+1} = \theta_2 z_2^t + w_2 \tag{24}$$

where $w_1, w_2 \sim \mathcal{N}(0, \sigma^2)$ are two independent random variables with normal distribution and $\theta_1, \theta_2 \in [0, 1]$ are two arbitrary parameters defining the autoregressive process. We then use the graphs $g$ to obtain coherent tuples, $z_t$. We built a dataset of $T = 10^4$ samples $\mathcal{D} = \{z_t\}_{t=1}^T$, applied a random reshuffle and trained a LightGBM regressor to predict $z_{t+1}$ from $z_t$ on 10% of $\mathcal{D}$. At each time $t$, we obtained a predictive distribution in terms of samples $\{(\tilde{z}_{t,s})_{s=1}^S\}$ with $S = 200$ via bootstrap on a calibration set with 40% of the $\mathcal{D}$, and we used the remaining 50% to test the theorems. Figure 5 shows a hypersurface (paraboloid) and a manifold with codimension $>1$ (quadratic-exponential) as examples of $M$. On both panels, three examples of predictions in terms of deterministic points $\hat{z}$, associated bootstrapped samples, their projection onto $M$, and true points $z$ are shown.

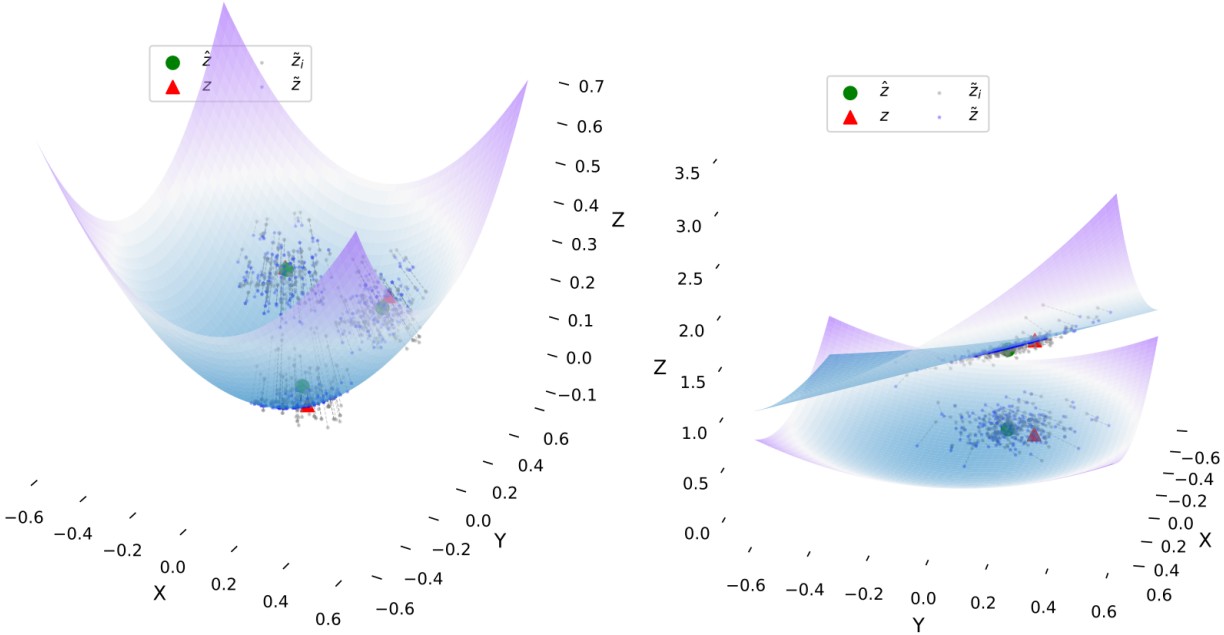

Figure 5: Examples of test cases in terms of tuples of predictions (green dots), ground truth (red triangles), and samples from predictive and reconciled distributions (gray and blue dots). Left: paraboloid case, Right: vector-valued function. In this case, just one tuple is plotted, but two points are visible since $f : \mathbb{R}^4 \to \mathbb{R}^2$.

For Theorems 1 and 2, we retrieved the number of predicted RMSE reduction cases and the false positives (cases in which the theorems predict a decrease of RMSE while observing an increase). We anticipate that we observed a total of 0 false positives across all the functions.

Since equation 17 provides a probabilistic estimation, we assess it with a randomized study. We generate 25 studies with associated datasets $\mathcal{D}_r$ of $10^4$ points from $M$ using the data generation process equation 24, with random parameters $\theta_1$ and $\theta_2$. We follow the previously described procedure to obtain predictive distributions via bootstrap on each test set. For each point on the test set we then evaluate 17, $\tilde{e}_{mc,t}$. To estimate the probability of RMSE reduction defined in equation 18, we adopted a binned calibration

estimator. We used a fixed width of $\Delta = 0.01$ and estimated $\pi(e_t)$ as

$$\hat{\pi}(e_t) = \frac{1}{N} \sum_{i \in \mathcal{A}_\Delta^N(e_t)} y_i \tag{25}$$

where $\mathcal{A}_\Delta^N(e_t)$ is the binned archive to which $e_t$ belongs. As before, $y$ is the binary outcome for the ex-post condition (i.e., whether reconciliation led to a lower RMSE after the uncertainty resolved).

Since we equip the calibration estimation with upper and lower boundaries via Clopper-Pearson intervals, we also test for the binary coverage of the intervals, defined as:

$$c = \sum_{i \in \mathcal{D}_{te}} c_i \quad c_i = \begin{cases} e_{L,i} > 0.5 & \text{if} \quad \|\tilde{\delta}\| < \|\hat{\delta}\| \\ e_{U,i} < 0.5 & \text{otherwise} \end{cases} \tag{26}$$

where $e_U, e_L$ are the upper and lower bounds defined in equation 20.

Finally, considering the case in which the forecasting task doesn't explicitly require $\hat{z} \in M$, we investigate if 3 can be used to craft an effective reconciliation strategy, reconciling just the points for which the estimated probability of a RMSE reduction is higher than a given threshold $\theta$:

$$z_t^\theta = \begin{cases} \tilde{z}_t := s(\hat{z}_t) & \text{if} \quad \tilde{e}_{mc,t} > \theta \\ \hat{z}_t & \text{otherwise} \end{cases} \tag{27}$$

The results are shown as the reduction in RMSE from the baseline RMSE $\|\hat{\delta}\|$ (never reconcile), normalized for the RMSE of the optimal strategy, where the forecast $\hat{z}$ is reconciled if this guarantees a reduction in RMSE:

$$z_t^* = \begin{cases} \tilde{z}_t := s(\hat{z}_t) & \text{if} \quad \|\hat{\delta}\| > \|\tilde{\delta}\| \\ \hat{z}_t & \text{otherwise} \end{cases} \tag{28}$$

That is, $z_t^*$ represents the best possible ex-post strategy, or equivalently, the optimal reconciliation strategy with perfect knowledge of the future. The final normalized score is:

$$\Delta_{rel,opt}^{RMSE} = \frac{\|\hat{\delta}\| - \|z_t^\theta - z_t\|}{\|\hat{\delta}\| - \|z_t^* - z_t\|} \tag{29}$$

## 4.2 Hypersurfaces

Plots of the tested hypersurfaces are shown in 12 in the appendix. Table 2 shows the ratio of cases with an observed RMSE reduction (first column), with a predicted reduction from Theorem 1 (second column), and of false positives, for different levels of noise affecting the data generation process described in equation 24. Across all tested hypersurfaces and noise levels, we observed zero false positives, providing strong empirical support for Theorem 1.

In Figure 6, two examples of calibration plots are shown for a constant sign curvature (paraboloid, left) and a non-constant sign curvature (Himmelblau, right). Blue points show the collected $1.25 \times 10^5$ (25 studies on the 50% test sets of the $10^4$ datasets) RMSE differences $\|\hat{\delta}\| - \|\tilde{\delta}\|$ before and after reconciliation. Lines show the moving averages $\hat{\pi}(e)$ from equation 25. The top and bottom panels refer to the upper and lower bounds obtained from Clopper-Pearson intervals, respectively, while the middle one shows $\hat{\pi}(e)$ for the nominal response of equation 17. A higher inter-study variance is expected in the central region of the plot.

Table 3 reports the binary coverage $c$ from 26. We see that the coverage tends to decrease with the noise $\sigma$, possibly due to the deterioration of the forecasts. If the predicted distribution $\tilde{\nu}$ is not representative of the true distribution $\nu$, we expect the test to be unreliable.

Figure 7 shows boxplots in terms of $\Delta_{rel,opt}^{RMSE}$ defined in equation 29. Each boxplot contains $\Delta_{rel,opt}^{RMSE}$ from the 25 randomized studies. From the definition of $\Delta_{rel,opt}^{RMSE}$, positive values indicate a reduction in RMSE compared to the unreconciled case, while a score of 1 represents the best binary reconciliation strategy.

|  |  | $\|\tilde{\delta}\| < \|\hat{\delta}\|$ | Th. 2 reduction | Th. 2 FP |
|---|---|---|---|---|
| $\sigma = 0.1$ | Abs | 0.59 | 0.13 | 0.00 |
|  | Exponential | 0.87 | 0.46 | 0.00 |
|  | MixedQuadratic | 0.37 | 0.15 | 0.00 |
|  | Paraboloid | 0.34 | 0.10 | 0.00 |
|  | Quartic | 0.34 | 0.11 | 0.00 |
| $\sigma = 0.3$ | Abs | 0.59 | 0.13 | 0.00 |
|  | Exponential | 0.68 | 0.36 | 0.00 |
|  | MixedQuadratic | 0.38 | 0.15 | 0.00 |
|  | Paraboloid | 0.33 | 0.10 | 0.00 |
|  | Quartic | 0.33 | 0.11 | 0.00 |
| $\sigma = 0.5$ | Abs | 0.59 | 0.13 | 0.00 |
|  | Exponential | 0.60 | 0.36 | 0.00 |
|  | MixedQuadratic | 0.38 | 0.15 | 0.00 |
|  | Paraboloid | 0.33 | 0.10 | 0.00 |
|  | Quartic | 0.30 | 0.12 | 0.00 |
| $\sigma = 0.7$ | Abs | 0.59 | 0.13 | 0.00 |
|  | Exponential | 0.55 | 0.33 | 0.00 |
|  | MixedQuadratic | 0.38 | 0.15 | 0.00 |
|  | Paraboloid | 0.33 | 0.10 | 0.00 |
|  | Quartic | 0.29 | 0.09 | 0.00 |
| $\sigma = 0.9$ | Abs | 0.59 | 0.13 | 0.00 |
|  | Exponential | 0.52 | 0.33 | 0.00 |
|  | MixedQuadratic | 0.39 | 0.15 | 0.00 |
|  | Paraboloid | 0.33 | 0.10 | 0.00 |
|  | Quartic | 0.28 | 0.09 | 0.00 |

Table 2: Predictions and false positives for Theorem 1 for different hypersurfaces with constant sign curvature, increasing levels of noise $\sigma$.

|  | $\sigma = 0.1$ | $\sigma = 0.3$ | $\sigma = 0.5$ | $\sigma = 0.7$ | $\sigma = 0.9$ |
|---|---|---|---|---|---|
| Rosenbrock | 0.59 | 0.52 | 0.49 | 0.47 | 0.45 |
| Ackley | 0.59 | 0.50 | 0.37 | 0.28 | 0.21 |
| Rastrigin | 0.63 | 0.51 | 0.40 | 0.31 | 0.25 |
| Quartic | 0.65 | 0.64 | 0.65 | 0.65 | 0.66 |
| Paraboloid | 0.67 | 0.67 | 0.67 | 0.66 | 0.66 |
| MixedQuadratic | 0.69 | 0.68 | 0.67 | 0.66 | 0.66 |
| Himmelblau | 0.75 | 0.68 | 0.63 | 0.57 | 0.54 |
| Abs | 0.81 | 0.81 | 0.81 | 0.81 | 0.81 |
| Exponential | 0.84 | 0.68 | 0.57 | 0.57 | 0.58 |

Table 3: Binary coverage from equation 26 for different hypersurfaces, increasing levels of noise $\sigma$.

The first boxplot represents the strategy in which the points are always reconciled, while the others represent the strategy of equation 27 with increasing values of $\theta$. For the exponential manifold, the best strategy, close to the optimal, is always reconciling. For the Rastring manifold, reconciling always worsens the RMSE, and choosing $\theta = 0.6$ only slightly improves the RMSE over the unreconciled case. For the other 7 tested hypersurfaces, reconciling improves the RMSE, and the best reconciliation strategy based on equation 17 uses a threshold of $\theta = 0.5$ or $\theta = 0.6$.

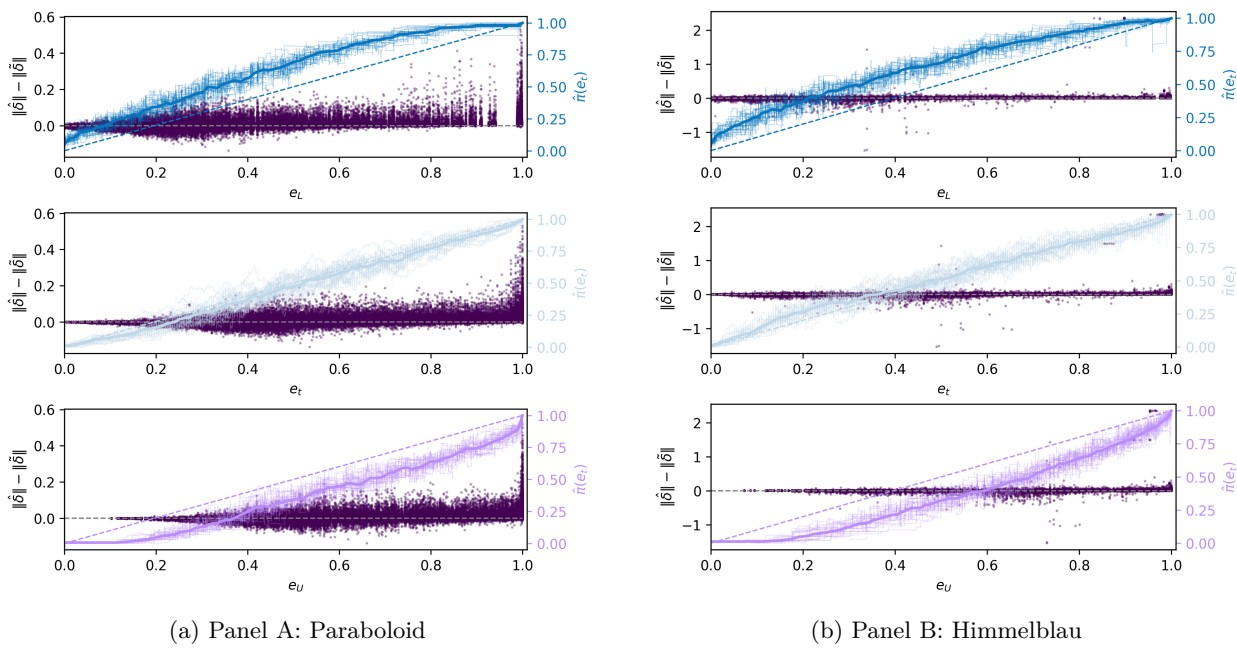

(a) Panel A: Paraboloid

(b) Panel B: Himmelblau

Figure 6: Calibration plots for equation 17 for two examples of constant sign curvature (left) and non-constant sign curvature (right) hypersurfaces, for noise level $\sigma = 0.3$.

### 4.3 Manifolds with codimension $> 1$

We repeat the same tests for manifolds with codimension>1, using Theorem 2 for manifolds with convex sub-level sets. Figure 13 in the appendix shows plots for the tested manifolds. Table 4 shows the ratio of cases with an observed RMSE reduction (first column), with a predicted reduction from Theorem 2 (second column), and of false positives for manifolds with convex sub-level sets. Also in this case, no false positives are present. This empirical finding is fully consistent with Theorem 2 prediction that false positives should never occur under its assumptions.

Figure 8 shows the calibration plots for a quadratic-quartic and for a bowl-sin manifold. Also in this case, for the manifold shown in the first panel, we can observe some underestimation of the probability of an improvement due to reconciliation in low-density zones of the plot (low probabilities). As before, this is likely due to the forecast $\tilde{\nu}$ being unreliable in these conditions.

The binary coverage is shown in Table 5, where the coverage is lower under high noise $\sigma$ for the ring-trig and bowl-sin manifolds, which are the most irregular ones. Also in this case, this is likely the effect of the forecast $\tilde{\nu}$ becoming unreliable.

Finally, Figure 9 shows boxplots in terms of $\Delta_{rel,opt}^{RMSE}$ for different manifolds. Also in this case, for the saddle-poly, exp-cosh, and Rosenbrock style manifolds, always reconciling the sample is close to the optimal strategy. For the other manifolds, as for the hypersurface tests, the best reconciliation strategy based on equation 17 uses a threshold of $\theta = 0.5$ or $\theta = 0.6$.

## 5 Note on geodesics and optimal deterministic predictions

This paper presented RMSE reduction conditions for nonlinear manifolds. However, for some forecasting tasks, other metrics could be more relevant. If we have a coherent forecast distribution $\tilde{\nu}$ supported on $M$, finding the best point forecast is not as straightforward as in the linear case. In general, given a sample-based

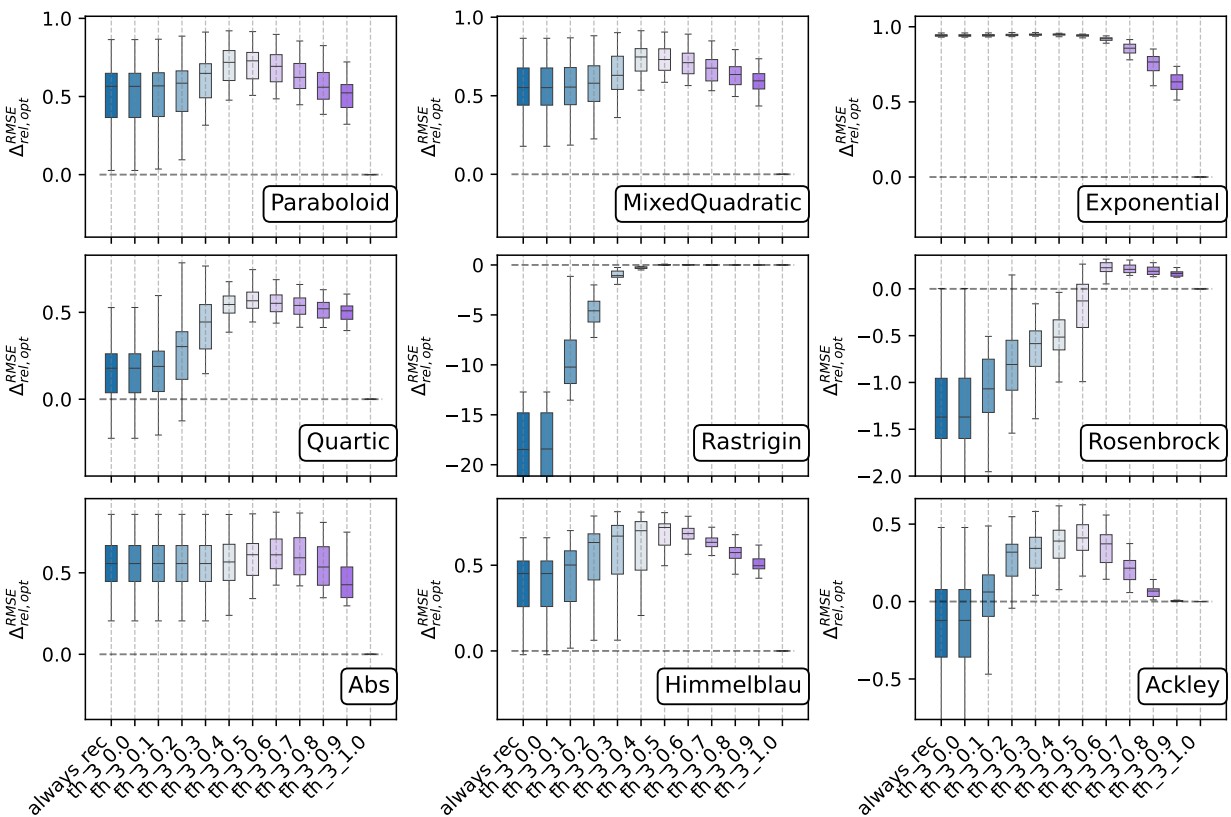

Figure 7: Boxplots of normalized improvements in RMSE for different reconciliation strategies for hypersurfaces.

representation of $\tilde{\nu}$, $(\tilde{z}_i, \tilde{\pi}_i)_{i=1}^{S}$, the best deterministic point can be found solving:

$$z^* = \arg\min_{z \in M} \sum_{i=1}^{S} \tilde{\pi}_i l(z, \tilde{z}_i)^2 \tag{30}$$

In the linear case $M = \mathbb{R}^n$ and under RMSE loss $l$, the minimizer is the weighted average of the samples, while when $M$ is a nonlinear manifold, this must be solved numerically. For general Manifolds, problem 30 is also known as Fréchet mean (Bačák, 2014; Lou et al., 2020), and is in general non-trivial to solve. For some forecasting tasks, it may be more meaningful to consider the geodesic distance $g$ as the loss $l$, rather than the RMSE. Defined $\gamma : [a, b] \to M$ a curve on the manifold, the geodesic between $a$ and $b$ is the distance of the shortest path on $M$ between these two points:

$$g(a, b) = \inf \int_a^b \|\gamma'(t)\|_\rho \, dt \tag{31}$$

where the Riemannian metric $\rho = (\rho_x)_{x \in M}$ is a smooth collection of inner products $\rho_x : T_x M \times T_x M \to \mathbb{R}$. The following example illustrates the importance of considering geodesic distance for some forecasting tasks.

|  |  | $\|\tilde{\delta}\| < \|\hat{\delta}\|$ | Th. 2 reduction | Th. 2 FP |
|---|---|---|---|---|
| $\sigma = 0.1$ | 4d-4d | 0.36 | 0.09 | 0.00 |
|  | bowl-sin | 0.33 | 0.10 | 0.00 |
|  | e2-s2 | 0.34 | 0.10 | 0.00 |
|  | quad-qq | 0.39 | 0.04 | 0.00 |
|  | quad-quad | 0.33 | 0.10 | 0.00 |
| $\sigma = 0.3$ | 4d-4d | 0.35 | 0.09 | 0.00 |
|  | bowl-sin | 0.36 | 0.07 | 0.00 |
|  | e2-s2 | 0.32 | 0.09 | 0.00 |
|  | quad-qq | 0.39 | 0.02 | 0.00 |
|  | quad-quad | 0.33 | 0.10 | 0.00 |
| $\sigma = 0.5$ | 4d-4d | 0.32 | 0.09 | 0.00 |
|  | e2-s2 | 0.29 | 0.08 | 0.00 |
|  | quad-qq | 0.42 | 0.01 | 0.00 |
|  | quad-quad | 0.33 | 0.10 | 0.00 |
| $\sigma = 0.7$ | 4d-4d | 0.27 | 0.09 | 0.00 |
|  | e2-s2 | 0.28 | 0.07 | 0.00 |
|  | quad-qq | 0.48 | 0.01 | 0.00 |
|  | quad-quad | 0.33 | 0.10 | 0.00 |
| $\sigma = 0.9$ | 4d-4d | 0.25 | 0.09 | 0.00 |
|  | e2-s2 | 0.27 | 0.06 | 0.00 |
|  | quad-qq | 0.52 | 0.00 | 0.00 |
|  | quad-quad | 0.33 | 0.10 | 0.00 |

Table 4: Predictions and false positives for Theorem 2 for different $M$ with codimension>1 and convex sub-level sets, increasing levels of noise $\sigma$.

|  | $\sigma = 0.1$ | $\sigma = 0.3$ | $\sigma = 0.5$ | $\sigma = 0.7$ | $\sigma = 0.9$ |
|---|---|---|---|---|---|
| quad-qq | 0.61 | 0.64 | 0.62 | 0.61 | 0.59 |
| ring-trig | 0.61 | 0.23 | 0.11 | 0.10 | 0.07 |
| e2-s2 | 0.64 | 0.66 | 0.69 | 0.71 | 0.71 |
| 4d-4d | 0.64 | 0.66 | 0.70 | 0.71 | 0.72 |
| exp-cosh | 0.64 | 0.64 | 0.64 | 0.62 | 0.59 |
| quad-quad | 0.65 | 0.64 | 0.63 | 0.64 | 0.63 |
| bowl-sin | 0.65 | 0.46 | 0.37 | 0.30 | 0.25 |
| saddle-poly | 0.69 | 0.70 | 0.70 | 0.67 | 0.65 |
| rosenbrock | 0.82 | 0.68 | 0.60 | 0.54 | 0.54 |

Table 5: Binary coverage for equation equation 17 for different $M$ with codimension>1, increasing levels of noise $\sigma$.

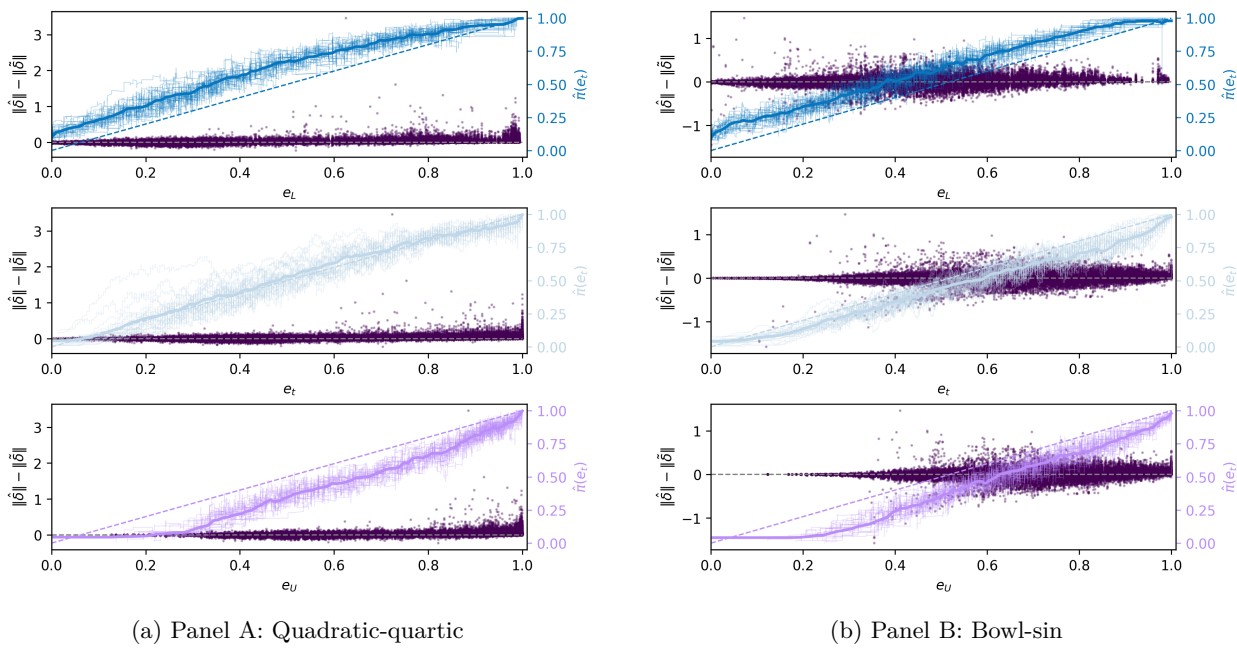

(a) Panel A: Quadratic-quartic  (b) Panel B: Bowl-sin

Figure 8: Calibration plots for equation equation 17 for two examples of $M$ with codimension $>1$ and convex (left) and non-convex (right) sub-level sets, for noise level $\sigma = 0.3$.

---

**Example 2: Optimal forecasts under geodesic distance**

*A heavy snowfall is forecast for tomorrow, with spatial probability distribution $\nu(z = [x, y])$, over the area surrounding a large lake. The event is expected to significantly disrupt road traffic. To reduce the economic impact, road maintenance crews can be preemptively deployed to a specific location along the road network, enabling a rapid response. Figure 10 illustrates two candidate deployment locations, $\tilde{z}_1$ and $\tilde{z}_2$, marked as green dots and selected by the decision maker. While $\tilde{z}_2$ is closer to the true disruption point $z$ (red triangle) in terms of Euclidean distance, deploying crews there would incur much higher operational costs than choosing $\tilde{z}_1$. This is due to the road network constraints, as shown by the red and green geodesic paths, which represent the cost of relocating crews from each candidate point to $z$ along the road infrastructure.*

---

## 6 Conclusions

This paper has advanced the theory of nonlinear forecast reconciliation through a geometric lens, establishing both deterministic guarantees and a practical probabilistic framework. Our first key contribution lies in Theorems 1 and 2, which provide sufficient conditions for strict RMSE reduction when reconciling forecasts via orthogonal projection onto constraint manifolds $M = \{z : f(z) = 0\}$. These deterministic results hold for codimension 1 and higher when all component functions $f_i$ exhibit convex sub-level or super-level sets.

However, our simulation study under isotropic-Gaussian errors revealed significant practical limitations: these geometric conditions apply only to subsets of prediction points, particularly on highly curved manifolds. This observation motivated our second major contribution— Theorem 3—which addresses the fundamental question of when reconciliation is likely to help. By introducing a consistent Monte Carlo estimator for $\Pr(\|z - \hat{z}\| \geq \|z - \tilde{z}\|)$, this probabilistic framework accommodates arbitrary manifolds without curvature assumptions or codimension restrictions, while providing confidence intervals.

Empirical validation confirmed the soundness of all theoretical contributions. Crucially, we observed zero false positives across all tests of Theorems 1 and 2, consistent with their theoretical guarantees. The estimator

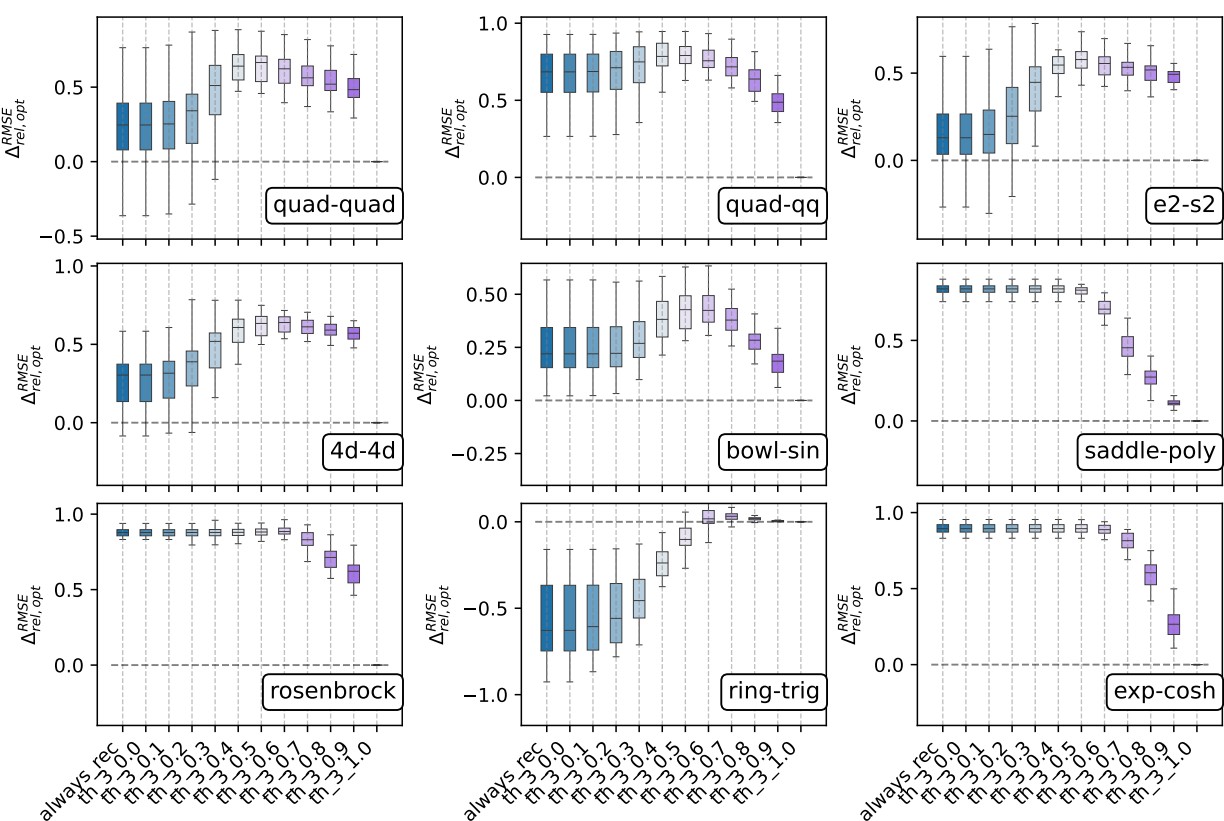

Figure 9: Boxplots of normalized improvements in RMSE for different reconciliation strategies for manifolds of codimension>1.

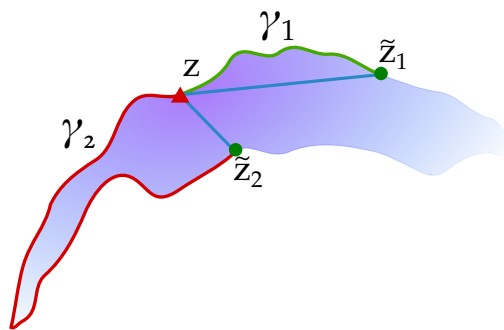

Figure 10: Conceptual plot for the importance of using geodesics as scoring rules in forecasting. Red triangle: true point $z$; green dots: examples of reconciled forecasts $\tilde{z}_1, \tilde{z}_2$; blue lines: euclidean distances; red and green lines: shortest paths on $M$ from predictions to $z$.

from equation 17 demonstrated excellent calibration, with empirical coverage of confidence bands aligning closely with nominal levels. Most significantly, when using this probability estimate to decide whether to reconcile—rather than naïve always/never approaches—we achieved measurable RMSE reduction across diverse manifolds.

Collectively, these results bridge geometric theory with forecasting practice: while Theorems 1 and 2 establish fundamental error-reduction principles under idealized conditions, equation 17 and Theorem 3 deliver a versatile decision-making tool for real-world applications. This dual perspective enables more reliable forecast integration where theoretical guarantees meet practical uncertainty.

### 6.1 Future research directions

This paper lays the foundations for future research directions in the field of nonlinear reconciliation. An interesting research direction is to craft analogous theorems using the geodesic distance instead of the RMSE. An interesting research topic is to find a function of the geodesic that is a proper scoring rule for some classes of nonlinear manifolds. For example, the energy score is a proper scoring rule and it's a function of the Euclidean distance. Another interesting research direction is to find robust rules to set the threshold parameter $\theta$ that maximizes the accuracy of the reconciliation strategy. Lastly, the RMSE reduction theorems and associated reconciliation strategies could be applied to nonlinear state estimation problems.

### Acknowledgments

This work has been funded by the Swiss State Secretariat for Education, Research and Innovation (SERI) under the Swiss contribution to the Horizon Europe projects DR-RISE (Horizon Europe, Grant Agreement No. 101104154) and REEFLEX (Horizon Europe, Grant Agreement No. 101096192). The authors would like to thank Lorenzo Zambon for his insightful feedback and valuable suggestions.

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

# A  Appendix

## A.1  Equivalence of problem equation 1 and FASE method

The FASE method (Brown Do Coutto Filho & De Souza, 2009) formulates the state estimation procedure using a discrete time-variant state-space representation:

$$x_{k+1} = F_k x_k + g_k + w_k \tag{32}$$

$$y_k = h_k\left(x_k\right) + v_k \tag{33}$$

Usually, in FASE, the state $x$ are the magnitudes and angles of the voltages, while $y$ represents the active and reactive powers. Equation 33 represents an a-priori forecast of the state $x$ at time $t+1$ given the state estimation at time $t$. Nonlinear forecasting models could also be used, leading to variations of the method, but linear models have been found to be difficult to beat.

The objective of the FASE method is formulated as:

$$\min_{x} \|\hat{y} - h(x)\|_R^2 + \|x - \hat{x}\|_M^2 \tag{34}$$

where $R$ and $M$ are positive definite weighting matrices for the forecast errors in $y$ and $x$, respectively.

To turn the nonlinear projection problem of 1 into the FASE problem of 34, we define $z = [x^T, y^T]^T$, $M = \{f(z) = 0\} = \{y = h(x)\}$. Then 1 becomes:

$$
\begin{aligned}
\min_{z} \quad & \|z - \hat{z}\|_W^2 \\
\text{s.t.} \quad & y = h(x)
\end{aligned}
\tag{35}
$$

with weight matrix

$$
W = \begin{bmatrix} M & 0 \\ 0 & R \end{bmatrix}
$$

The constraint $y = h(x)$ implies that any feasible point $z$ must satisfy

$$
z = \begin{bmatrix} x \\ h(x) \end{bmatrix}
$$

We can therefore rewrite the projection problem as an unconstrained optimization over $x$:

$$
\min_{x} \left\| \begin{bmatrix} x \\ h(x) \end{bmatrix} - \begin{bmatrix} \hat{x} \\ \hat{y} \end{bmatrix} \right\|_W^2
$$

Expanding the norm with block-diagonal weights:

$$
\left\| \begin{bmatrix} x - \hat{x} \\ h(x) - \hat{y} \end{bmatrix} \right\|_W^2 = \|x - \hat{x}\|_M^2 + \|h(x) - \hat{y}\|_R^2
$$

This is exactly the objective function of the FASE formulation. This shows the two objectives are equivalent when considering a graph $y = h(x)$ for the definition of the manifold and when the weight matrix $W$ is block-diagonal, partitioned according to the dimensionality of $x$ and $y$.

### A.2   Second fundamental form

Given $f : \mathbb{R}^n \to \mathbb{R} \in C^2$ , $M = \{z : f(z) = c\} \subset \mathbb{R}^n$ is a level set of $f$ and a regular hypersurface of $f$ if $\nabla f(z) \neq 0$ on $M$. $H = D^2 f(z)$ is the Hessian of the function at $z$ , $E_z \in \mathbb{R}^{n \times (n-1)}$ an orthonormal basis of the tangent space $T_x M$ at $x$. Then the *second fundamental form* $\mathbb{I}_p \in \mathbb{R}$ at a point $p \in M$ is a symmetric bilinear form on the tangent space $T_p M$, defined by:

$$
\mathbb{I}_p(z, w) = \langle D_z \nu, w \rangle, \quad z, w \in T_p M
\tag{36}
$$

$$
\nu(z) = \frac{\nabla f(z)}{\|\nabla f(z)\|}
\tag{37}
$$

where $\nu(z)$ is the unit normal vector field, measures the rate of change of the normal vector in the direction $z$, i.e., the normal curvature. In coordinates, this reduces to:

$$
\mathbb{I}_p(z, z) = \frac{z^\top H(p) z}{\|\nabla f(p)\|}, \quad z \in T_p M
\tag{38}
$$

which describes the change of the normal acceleration of the surface in the direction of $x$ belonging to the tangent space. A sub-level set is locally convex iff the second fundamental form is positive semidefinite (theorem 1 and 3 in Thorpe (1979)):

$$
z^\top H(p) z \geq 0 \quad \text{for all } z \in T_z M
\tag{39}
$$

We can test this condition by inspecting if the Hessian projected on the tangent space is positive definite, or equivalently, by checking if the smallest eigenvalue of the restricted Hessian $H_{\tan}(p)$ is greater than 0:

$$H_{\tan}(p) = E_p^\top H(p) E_p \tag{40}$$

$$\lambda_{\min}(H_{tan}(p)) > 0 \Leftrightarrow H_{\tan}(p) \succ 0 \tag{41}$$

where $E_p \in \mathbb{R}^{n \times (n-1)}$ is an orthonormal basis of the tangent space $T_p M$ at $p$. This shows that $\lambda_{\min}(H_{tan}(p))$ is a natural choice for inspecting the curvature of the levelset at $p$. Figure 11 shows different level sets for $f(z) = z_1^4 - 3z_1^2 + z_2^2$ (left axis) and a specific level set for $f(z) = z_1^4 - z_1^2 + z_2^2 + z_3^2 - 2z_1 z_3$ (right axis), both colored by $\lambda_{min}(H_{tan})$. The second fundamental form can be used to write the following second-order approximation for $M$:

$$z = \tilde{z} + t + \frac{1}{2}\nu(\tilde{z})\underbrace{\mathbb{I}_{\tilde{z}}(t,t)}_{\text{scalar}} + o\left(\|t\|^2\right) \tag{42}$$

where $t \in T_{\tilde{z}}M$ is in the tangent direction, while the correction introduced by the second fundamental form lies in the direction of the unit normal $\nu(\tilde{z})$, $\mathrm{Im}(\nabla f) \subset \mathbb{R}^n$.

### A.2.1 Vector valued functions

In the case of manifold with codimension higher than 1, that is defined by a vector valued constraint $f : \mathbb{R}^n \to \mathbb{R}^m$, $M = \{z \in \mathbb{R}^n : f(z) = 0\}$, $m < n$, the normal space $\mathrm{Im}(J_f(z))$ spans the Jacobian rows. In this case $\mathbb{I}_{\tilde{z}}(t,t) \in \mathbb{R}^m$ is a vector and the analogous expression to equation 42 for second order approximation can be written as:

$$z = \tilde{z} + t + \frac{1}{2}J_f(\tilde{z})^\top \cdot \mathbb{I}_{\tilde{z}}(t,t) + o\left(\|t\|^2\right) \tag{43}$$

For the vector-valued case, an orthonormal basis $E$ for the tangent space $T_z M = \ker J(z)$ can be found by QR decomposition of the transposed Jacobian.

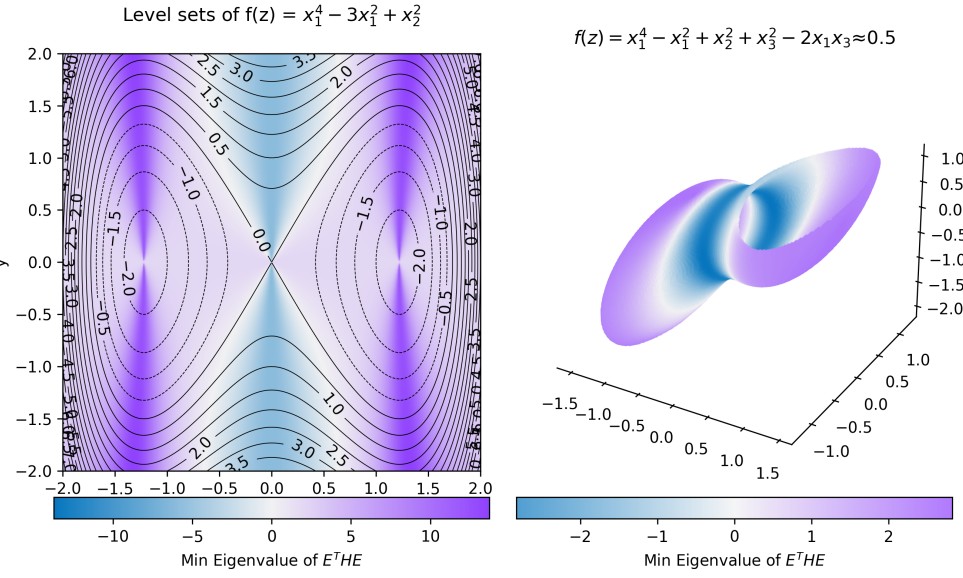

Figure 11: Two examples of hypersurfaces colored by $\lambda_{min}(H_{tan})$. These examples illustrate how $\lambda_{min}(H_{tan})$ can be thought of as the degree with which the surface *bends away* or *bends towards* a point in the ambient space.

### A.3 Proof of equation 12

In the following, we prove equation 12 for hypersurface $M = \{z : f(z) = 0\} \subset \mathbb{R}^n$ with $f$ having a convex sub or super level set. Consider two points $z$ and $\tilde{z}$, both belonging to $M$ and minimally displaced $\varepsilon = z - \tilde{z}$. We can write the second-order Taylor expansion at $\tilde{z}$:

$$0 = f(z) = f(\tilde{z} + \varepsilon) = f(\tilde{z}) + \nabla f(\tilde{z})^T \varepsilon + \frac{1}{2}\varepsilon^T H(\tilde{z})\varepsilon + o\left(\|\varepsilon\|^2\right) \tag{44}$$

Since $f(\tilde{z}) = 0$, and since $\varepsilon$ becomes tangent to $M$ in the limit, neglecting higher order terms, we can write:

$$\nabla f(\tilde{z})^T \varepsilon = -\frac{1}{2}\varepsilon^T H(\tilde{z})\varepsilon \quad \forall \varepsilon \in T_{\tilde{z}}M \tag{45}$$

where $T_{\tilde{z}}M$ is the tangent space at $\tilde{z}$. Equivalently, we can write:

$$\nabla f(\tilde{z})^T \varepsilon = -\frac{1}{2}E_{\tilde{z}}^T H(\tilde{z})E_{\tilde{z}} \tag{46}$$

where $E_{\tilde{z}} \in \mathbb{R}^{n \times (n-1)}$ is an orthonormal basis of the tangent space $T_{\tilde{z}}M$ at $\tilde{z}$. Then:

$$\text{sign}\left(\nabla f(\tilde{z})^T \varepsilon\right) = -\text{sign}\left(E_{\tilde{z}}^T H(\tilde{z})E_{\tilde{z}}\right) = -\text{sign}\left(\lambda_{min}(H_{tan}(\tilde{z}))\right) \tag{47}$$

where the last equality comes from the definiteness of the restricted Hessian for convex / concave curves equation 39-equation 41. We proved equation 47 in the limit $\varepsilon \to 0$, but since $M$ is the boundary of a convex set, it has a supporting hyperplane in $\tilde{z}$, which means that $\text{sign}\left(\nabla f(\tilde{z})^T \varepsilon\right)$ is constant on $M$, and thus equation 47 holds also for $\varepsilon = \tilde{\delta}$.

### A.4 On the vacuousness of instantaneous bounds for $\tilde{e} - Y_t$

We want to bound the error $|e_{mc,t} - \tilde{e}_t|$, that is, the distance from the estimated probability of improving RMSE after projection and the true probability or RMSE reduction. This can be decomposed via triangle inequality to consider the error of the Monte Carlo sampling:

$$|e_t - \tilde{e}_{mc,t}| \leq \underbrace{|e_t - \tilde{e}_t|}_{T} + \underbrace{|\tilde{e}_t - \tilde{e}_{mc,t}|}_{M}$$

The Monte Carlo estimation error $M$ can be bounded easily as shown in lemma 1.

Since $\nu_t$ is never observed, bounding $T$ is more challenging. We can try to make explicit the dependence of $T$ on the only observed sample $Z_t \sim \nu_t$ drawn from the true distribution at time $t$, using the decomposition:

$$|e_t - \tilde{e}_t| \leq \underbrace{|e_t - Y_t|}_{A} + \underbrace{|\tilde{e}_t - Y_t|}_{E} \tag{48}$$

where $Y_t = h(\phi_t(Z_t))$ is an indicator random variable, the evaluation of $h(\phi_t(\cdot))$ at the observed sample at time $t$. This decomposition has the following interpretation: $A$ represents the aleatoric uncertainty due to the unobservable distribution $\nu_t$, while $E$ represents the epistemic uncertainty, which can be made smaller by increasing the accuracy of the predicted law $\tilde{\nu}_t$. In the following we consider instantaneous bounds on these two quantities.

- $A$: Unless $\nu_t$ produces a degenerate distribution under $h_t(\phi_t(z_t))$, the best possible global, deterministic instantaneous bound for $A$ is $A < 1$. That is, unless the true distribution is totally contained in one of the two half-spaces defined by $\phi_t$, the best instantaneous bound on $A$ is uninformative since $e_t$ is a probability, bounded in 0, 1. A conditional (on the value of $e_t$) deterministic instantaneous bound is $\max(1 - e_t, e_t) \geq 1/2$. Furthermore, this bound approaches 1 when $e_t$ is close to 0 or 1, making it particularly uninformative.

- $E$: The best possible instantaneous bound on $E$ is also uninformative, since it is the one that can be obtained when $\tilde{e} = e_t$, that is, the bound under perfect knowledge of the true distribution $\nu_t$, which is again $\max(1 - e_t, e_t)$.

Summing the two components results in a totally uninformative bound $2\max(1 - e_t, e_t) \geq 1$ for $|e_t - \tilde{e}_t|$, therefore this specific decomposition can't be used to bound $T$ non-vacuously.

---

**Algorithm 1:** Augmented Lagrangian nonlinear reconciliation with L–BFGS line search

---

**Input:** forecast $\hat{z} \in \mathbb{R}^n$, SPD weight $W \succ 0$, constraint $f : \mathbb{R}^n \to \mathbb{R}^m$
**Input:** tolerances $tol_{\mathrm{feas}}, tol_{\mathrm{grad}}, tol_{\mathrm{step}}$, penalty parameters $\rho_0, \rho_{\mathrm{mult}}, \tau_{\mathrm{inc}}$
**Input:** max iterations $\mathrm{max\_outer}, \mathrm{max\_inner}$
**Whitening step:** compute Cholesky $W = L^\top L$ and set $L^{-1}$.
Set $\hat{y} \leftarrow L\hat{z}$.
Initialize $y^0 \leftarrow \hat{y}$, $\lambda^0 \leftarrow 0 \in \mathbb{R}^m$, $\rho^0 \leftarrow \rho_0$.
**for** $k = 0, 1, 2, \ldots, \mathrm{max\_outer} - 1$ **do**

    **Inner problem (unconstrained):**

$$\min_{y \in \mathbb{R}^n} \mathcal{L}_k(y) := \tfrac{1}{2}\|y - \hat{y}\|_2^2 + (\lambda^k)^\top c(y) + \tfrac{1}{2}\rho^k \|c(y)\|_2^2, \quad c(y) := f(L^{-1}y).$$

    Initialize $y^{k,0} \leftarrow y^k$ and L–BFGS state.
    **for** $j = 0, 1, 2, \ldots, \mathrm{max\_inner} - 1$ **do**
        Compute value and gradient $v^{k,j} \leftarrow \mathcal{L}_k(y^{k,j})$, $g^{k,j} \leftarrow \nabla \mathcal{L}_k(y^{k,j})$.
        Use a zoom line search (satisfying Wolfe conditions) to compute an L–BFGS step $\Delta y^{k,j}$ and
          update $y^{k,j+1} \leftarrow y^{k,j} + \Delta y^{k,j}$.
        **if** $\|g^{k,j}\|_\infty \leq tol_{\mathrm{grad}}$ **or** $\|\Delta y^{k,j}\|_2 \leq tol_{\mathrm{step}}$ **then**
            **break**

    Set $y^{k+1} \leftarrow y^{k,j_\star}$ (final inner iterate).
    Recover $z^{k+1} \leftarrow L^{-1}y^{k+1}$ and constraint $c^{k+1} \leftarrow f(z^{k+1})$.
    **if** $\|c^{k+1}\|_2 \leq tol_{\mathrm{feas}}$ **then**
         **break**                              `// outer stopping: feasibility reached`
    **Multiplier update:** $\lambda^{k+1} \leftarrow \lambda^k + \rho^k c^{k+1}$.
    **Penalty update:**

$$\rho^{k+1} \leftarrow \begin{cases} \rho^k \cdot \rho_{\mathrm{mult}}, & \text{if } \|c^{k+1}\|_2 > \tau_{\mathrm{inc}} \, tol_{\mathrm{feas}}, \\ \rho^k, & \text{otherwise.} \end{cases}$$

**return** $\tilde{z} = z^{k+1}$ *as the reconciled forecast.*

---

### A.5 ALM LBFGS solver

This pseudocode describes an L–BFGS method equipped with a zoom line search enforcing Wolfe conditions, starting from the previous iterate. We terminate the inner loop when the gradient norm and step size are below fixed tolerances, and then update the multipliers and penalty parameter in a standard Augmented Lagrangian Method (ALM) fashion.

## A.6   Plots of tested manifolds

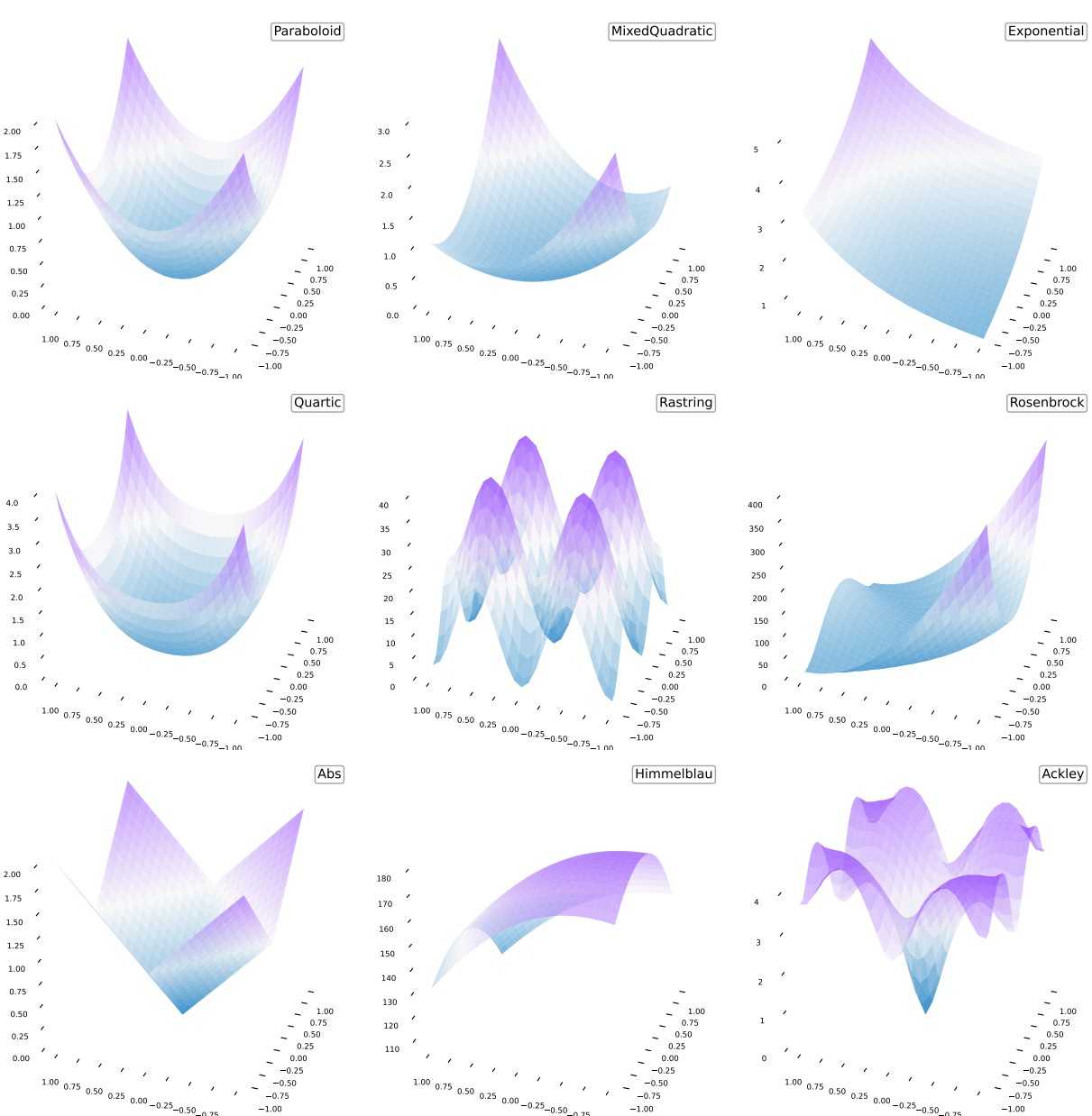

Figure 12: Plots of tested hypersurfaces.

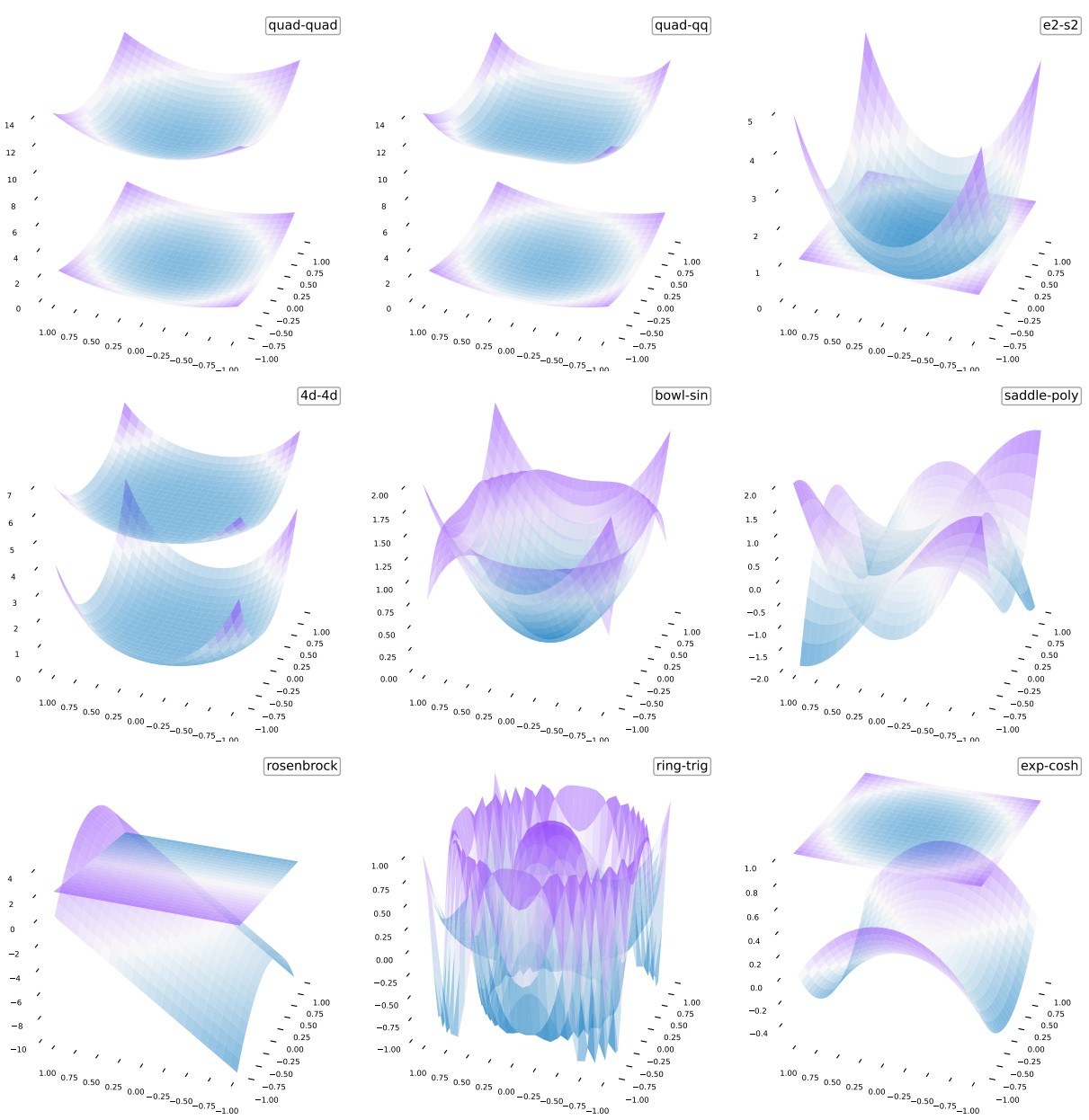

Figure 13: Plots of tested manifolds with codimension $> 1$. Since it's hard to visualize level sets of $f : \mathbb{R}^4 \to \mathbb{R}^2$, we plotted level sets of $f_1$ and $f_2$ separately.

