# OpenReview forum: "Nonlinear reconciliation: Error reduction theorems"
_TMLR — Accepted by TMLR_

### Review · Reviewer_QAzm · 2025-09-10

**Summary Of Contributions:**

The paper studies nonlinear forecast reconciliation by projecting base forecasts onto a constraint manifold. It gives sufficient error-reduction conditions for (i) hypersurfaces with constant-sign curvature and (ii) higher-codimension manifolds, and proposes a Monte-Carlo “probabilistic test” driven by a reconciled predictive distribution. A Newton/KKT solver is outlined, and synthetic experiments illustrate when reconciliation helps.

Strengths:
- clear geometric setup
- simple sufficient conditions
- a practical Monte-Carlo test
- simulations cover several toy manifolds.

Weaknesses:
- “probabilistic guarantees” amount to consistency of a plug-in estimator (no finite-sample/statistical guarantee under the true law)
- the optimization block appears without convergence or existence/uniqueness conditions
- empirical validation is synthetic only
- some figures are hard to read.

**Audience:**

Yes

**Audience Explanation:**

Forecast reconciliation is widely used, and extending linear results to nonlinear constraints is of interest to forecasting and constrained learning communities. The geometric conditions and the simple decision rule are likely to attract readers, even if the empirical scope and guarantees need strengthening.

**Claims And Evidence:**

No

**Claims Explanation:**

– The deterministic theorems are clean sufficient conditions, but only in the constant-curvature case, which is rather straightforward. The article fails to deliver meaningful results in the difficult cases.

– The “probabilistic” part is a Monte-Carlo estimator under the modelled forecast law; the guarantee given is SLLN consistency, not a bound on expected error reduction under the true data-generating distribution.

– Experiments are entirely synthetic with limited baselines; there is no real-data study to back practical relevance.

**Requested Changes:**

Critical :

1) Calibrate the claims around “probabilistic guarantees.” Say clearly that the test estimates the probability of improvement under the forecast law; separate modelling assumptions from statistical ones.
2) Add finite-sample bounds for the Monte-Carlo estimator (e.g., Hoeffding/Bernstein) and a simple sample-size guide for a target margin/CI.
3) State well-posedness and solver guarantees: conditions for existence/uniqueness of the orthogonal projection and local/global convergence of the Newton/KKT updates.
4) Clarify the role of W in Eq. (1). If results require W=I, say so; otherwise sketch the extension or move W≻0 to discussion.
5) Fix structure: relocate §2.1 next to the probabilistic test or add forward references so the flow is coherent.

Non-critical :

6) Improve figures.
7) Add at least one real-data benchmark.

---

> ### Author Response · Authors · 2025-11-25
> **Response to Reviewer QAzm**
>
> We deeply thank the reviewer for their valuable comments, we think we substantially improved the manuscript thanks to their feedback.
>
> > The deterministic theorems are clean sufficient conditions, but only in the constant-curvature case, which is rather straightforward. The article fails to deliver meaningful results in the difficult cases. The “probabilistic” part is a Monte-Carlo estimator under the modelled forecast law; the guarantee given is SLLN consistency, not a bound on expected error reduction under the true data-generating distribution.
>
> We agree that the previous theorem was straightforward and it's now turned into a lemma. Under additional assumptions we now provide a stronger probabilistic theorem with respect to the true law. This is now directly bounding the distance between the Monte Carlo estimator for the probability of RMSE reduction, and the true probability under true law, at time t. This theorem now provides a guidance for the difficult cases.
>
> > Experiments are entirely synthetic with limited baselines; there is no real-data study to back practical relevance.
>
> We are currently working on an applied paper in which we apply this technique on power flow equations under time-varying constraints, but we would like to try to publish this in a separate, more applied journal.
>
> >Calibrate the claims around “probabilistic guarantees.” Say clearly that the test estimates the probability of improvement under the forecast law; separate modelling assumptions from statistical ones.
>
> We have revised Theorem 3 to clarify its claims and assumptions, and to provide precise probabilistic guarantees on the probability of RMSE reduction after reconciliation. We explicitly separate modeling assumptions (e.g., curvature, forecaster class) from statistical assumptions (stationarity, i.i.d. archive, and forecaster reliability). Clopper–Pearson finite-sample bounds are provided for the probability that the reconciliation process reduces the RMSE, under a process stationarity assumption and using archive
> $A_{\Delta}= \{(e_i, y_i)\}_{t=1}^N$
>
> The archive comprises $N$ indicator functions (1 for successful reconciliation/error reduction), with predicted probabilities of improvement binned at level $e_i\in[e-\Delta, e +\Delta]$. We discuss the effect of the sample size $N$ on the reliability bounds:
> \begin{equation}
>         e_L(\beta,N_\Delta(e),k_\Delta(e)) \le \pi(e) \le e_U(\beta,N_\Delta(e),k_\Delta(e)).
> \end{equation}
> where $k_\Delta(e)$ is the number of successful reconciliations trials out of $N$ and $1-\beta $ is a desired confidence level. Here, the two–sided Clopper–Pearson lower bound is, $e_{L}(k,N;\beta)
> = \mathcal{B}^{-1}\bigl(\beta/2;k;N-k+1\bigr),
> $ where $\mathcal{B}^{-1}$ is the inverse beta function.
>
> >Add finite-sample bounds for the Monte-Carlo estimator (e.g., Hoeffding/Bernstein) and a simple sample-size guide for a target margin/CI.
>
> In the revised manuscript, we added an extended discussion on the practical use of Theorem 3 and guidance for data collection for a target margin/CI. For instance, given a confidence level $(1-\beta)$ and a target lower reliability $e_L$, e.g., at least 0.9, a minimal sample size $N^\star$ can be computed for a success rate $k/N$. This value can be obtained numerically by inverting the confidence interval. Although $k$ is not controllable (as it depends on the forecaster and geometry of the problem), this inversion provides a simple and practical guideline for experimental design and sample size selection.
>
>
>
> >State well-posedness and solver guarantees: conditions for existence/uniqueness of the orthogonal projection and local/global convergence of the Newton/KKT updates.
>
> The conditions of existence/uniqueness are now reported after the presentation of the algorithm. We do not derive new convergence results, but we rely on established theory. Furthermore, in the appendix is now present the pseudo-code for a new solver with improved convergence, which is also present in the python package. In our experiments, the algorithm converged robustly for all tested forecasts.
>
> >Clarify the role of W in Eq. (1). If results require W=I, say so; otherwise sketch the extension or move W≻0 to discussion.
>
> We dropped the sentence *in what follows we set \(W=I_{n}\), so the solution is the orthogonal projection of \(\hat{z}\) onto \(M\)* when we present the main algorithm, since the non-orthogonal projection method has been shown to be useful in linear reconciliation setting to improve predictions' accuracy. Orthogonality is a requirement for the first two theorems, but not for the estimation of $e$. We now clarify it after introducing the Monte Carlo estimator.
>
> >Fix structure: relocate §2.1 next to the probabilistic test or add forward references so the flow is coherent.
>
> We have now relocated the section
>
> > Improve figures.
>
> We have now added an additional figure explaining better the probabilistic estimator.

---

### Review · Reviewer_YjXa · 2025-09-22

**Summary Of Contributions:**

The paper considered the problem of whether reconciliation could reduce the prediction error under nonlinear constraints. Two sufficient conditions are proposed for the single constraint case and for multiple constraints, respectively, under the assumption of simple geometry (the sub/suplevel set is convex/concave) of the constrained manifold. For general constrained manifold, the authors give an estimate of the probability that projecting the prediction onto the constrained manifold will reduce the error.

**Additional Comments:**

NA

**Audience:**

Yes

**Audience Explanation:**

People who are working on forecast reconciliation may be interesed in this work.

**Claims And Evidence:**

Yes

**Claims Explanation:**

The results in this paper are supported by proofs.

**Requested Changes:**

Overall, the paper generalize the existing works on reconciliation under linear constraints to nonlinear constraints. However, the writing of the paper needs to be significantly improved. In particular, many notations are used without a clear definition. The detailed comments are as follows.

0. The author may further motivate the problem in the introduction. For example, why it is not a good choice to always do the projection since we already know the constraints?

1. Many words are vague, e.g., "nonlinear context" in the abstract, "linear case" at the beginning of section 3. I suggest replace them by nonlinear constraints and linear constraints.

2. The norm \|  \|_W should be defined.

3. Just below Eq. (1). Is "an (m-n)-dimensional differentiable manifold" an assumption?

4. The M defined for the linear reconciliation problem is not consistent in dimension.

5. Eq. (4). Is the linear system uniquely solvable?

6. Theorem 1. \lambda_min is not defined. What is the meaning of "restricted"? It would be better to present Theorem 1 in a local manner, i.e., state the assumptions locally near the exact value and the prediction.

7. Proo1. \tilde \delta is already defined and cannot be arbitrary. In addition, It is not clear what is the variable in "\nabla f(\tilde z)^T \tilde \delta is a constant on M".

8. Section 3.2. Vector-valued functions might be replaced with multiple constraints for clarity.

9. Section 3.3. What is the meaning of "at multiple noise level"? In addition, I think \sigma_I should be somehow related to the magnitude of z_i.

10. Example 1. M should be defined as M = {z = (x, y)^T \in \R^2: ...}. Is data D a subset of \R^2? What are z_{1, i} and z_{2, i}?

11. Theorem 3. k is not defined.

---

> ### Author Response · Authors · 2025-11-25
> **Reviewer YjXa**
>
> We thank the reviewer for their valuable comments.
> >The author may further motivate the problem in the introduction. For example, why it is not a good choice to always do the projection since we already know the constraints?
>
> If we are reconciling as a way to reduce RMSE, in the presence of nonlinear constraints, it's not always guaranteed that projecting will decrease the RMSE.
>
> *However, when the constraints are nonlinear, such a reduction is not always guaranteed: if reconciliation is performed solely to boost accuracy—rather than to strictly enforce the constraints—it can in fact lead to worse predictions. The ability to reliably estimate the probability of RMSE reduction under nonlinear constraints will significantly drive the adoption of reconciliation for improving forecasting accuracy.*
>
> > Many words are vague, e.g., "nonlinear context" in the abstract, "linear case" at the beginning of section 3. I suggest replace them by nonlinear constraints and linear constraints.
>
> We have changed the mentioned terms to "linear constraints" and "nonlinear constraints" in the manuscript.
>
> > The norm$ ||_W$ should be defined.
>
> We have defined the norm $||_W$ as - $\lVert z\rVert_W=z^TWz$
>
>
> > Just below Eq. (1). Is "an $(m-n)$-dimensional differentiable manifold" an assumption?
>
> This is indeed an assumption which we have mentioned below Eq. (1).
>
> > The $M$ defined for the linear reconciliation problem is not consistent in dimension.
>
> We revised as follows:  *where we assume $M:=\{z\in\mathbb{R}^{n}\mid f(z)=0\}$ where $f: \mathbb{R}^n \rightarrow \mathbb{R}^m$ is smooth and 0 is a regular value of $f$. Under this assumption, $M$ is an \(n-m\)-dimensional differentiable manifold, with codimension \(m\), embedded in $\mathbb{R}^{n}$*.
>
> We also clarified that the dimension of M refers to its intrinsic manifold dimension, independently of the ambient dimension in which it is embedded.
>
> > Eq. (4). Is the linear system uniquely solvable?
>
> The required condition is full rank of the Jacobian matrix at $z_k$. We've now added a new paragraph in which we discuss existence and uniqueness of the solution, convergence conditions and introduce a modification of the main algorithm that we've made available in the python package.
>
> > Theorem 1. $\lambda_{min}$ is not defined. What is the meaning of "restricted"? It would be better to present Theorem 1 in a local manner, i.e., state the assumptions locally near the exact value and the prediction.
>
> We have now defined it: *and $\lambda_{min}$ denotes its smallest eigenvalue*
> We have now better defined the restricted Hessian in the appendix; we've now put a reference directly in the body of the theorem
>
> Finally, we restated the theorem highlighting the fact that the condition can be computed locally:
>
> *A sufficient local condition, that can be computed from the value of $f$ at the forecasted point $\hat z$ and the Hessian at the projection point $\tilde z$, for the reduction of the RMSE after projecting $\hat{z}$ into $\tilde{z}$ is the following*
>
> > Proof1. $\tilde \delta$ is already defined and cannot be arbitrary. In addition, It is not clear what is the variable in "$\nabla f(\tilde z)^T \tilde \delta$ is a constant on M".
>
> The proof relies on the fact that
> *A bounding condition for \eqref{eq:reduction_2} to be positive is $\delta_{\pi}^T\tilde{\delta}\geq 0 \ \forall \ \tilde{\delta}$*
>
> that is, if we can show that this condition holds for arbitrary values of $\tilde{\delta}$ the theorem holds. We proceed in this way since $\tilde{\delta}$ depends on where the true point will be on the manifold, which is unknown at prediction time, so the need of bounding the expression for any value of $\tilde{\delta}$. For the same reason the variable in the sentence *the sign of $\nabla  f(\tilde{z})^T\tilde{\delta}$ is constant on $M$* is $\tilde\delta$. We now restate more clearly this way to construct the proof.
>
> > Section 3.2. Vector-valued functions might be replaced with multiple constraints for clarity.
>
> The section title has been revised as suggested
>
> > Section 3.3. What is the meaning of "at multiple noise level"? In addition, I think $\sigma_I$ should be somehow related to the magnitude of $z_i$.
>
> We now replaced "noise level" with "increasing noise standard deviations". By keeping the magnitude of the error fixed but increasing it, we are increasing the average distance of the points from the manifold. This was done to study what happens when errors are isotropic around the manifold, not to inject preferential directions for the error. Preferential directions would arise if the errors are proportional to $z_i$.
>
> > Example 1. M should be defined as $M = {z = (x, y)^T \in \R^2: ...}$. Is data $D$ a subset of $\R^2$? What are $z_{1, i}$ and $z_{2, i}$?
>
> There was indeed a slight abuse of notation. It is now fixed.
>
> > Theorem 3. k is not defined.
>
> Theorem 3 has been revised. The quantity $k$ is now defined as $k_\Delta(e) := \sum_{i\in \mathcal A_\Delta(e)} Y_i$

---

### Review · Reviewer_FCkM · 2025-11-11

**Summary Of Contributions:**

This research provides theoretical guarantees regarding whether nonlinear reconciliation, a method for transforming predictions to satisfy nonlinear constraints, actually reduces the Root Mean Squared Error (RMSE). Reconciliation is an important topic in practical situations where predictions are expected to satisfy appropriate constraints. However, there is not much research on the case with nonlinear constraints.

This study derives several theorems regarding the conditions under which reconciliation reduces the RMSE. First, Theorems 1 and 2 derive sufficient conditions for the RMSE to be improved by reconciliation for each given prediction point. However, subsequent experiments reveal that this condition is too optimistic. Therefore, Theorem 3 provides a method for numerically estimating the probability of improvement, rather than providing a binary indication of whether improvement occurs at each point. The usefulness and validity of the derived theorems are verified through detailed numerical experiments. This provides guidance on whether or not reconciliation should be performed.

**Audience:**

Yes

**Audience Explanation:**

Reconciliation is an important topic in machine learning and data analysis, and guaranteeing performance improvements under broader constraints is crucial. In this context, this work provides important insights related to reconciliation. Therefore, it is relevant to the readership of Transactions on Machine Learning Research.

**Claims And Evidence:**

Yes

**Claims Explanation:**

Basically, the paper is well written, and the claims are carefully examined by extensive numerical experiments. On the other hand, there are some concerns. I recommend making some improvements in this regard.

**Requested Changes:**

# major points
* My main concern is that without background knowledge on how reconciliation is used, the setup of Section 4. Numerical Tests will be hard to read. Theorems 1-3 abstractly deal with the effects of projecting onto a manifold, and can be understood independently. However, as soon as we get to Section 4.1, it becomes very specific, and readers may be confused by things like generating data using an AR model (Eq. 19) and performing training with LightGBM. Since TMLR readers cover a wide range of machine learning expertise, I think this is a major concern. At least, I was very confused when I first read it. Therefore, I recommend adding a paragraph or subsection in Section 4.1 that clearly explains what kind of data analysis problem is being addressed and how reconciliation is involved, in a way that is easy for those who have not had much experience with reconciliation to understand.

# minor points
* Regarding the notation in Theorem 3, it seems somewhat strange that $\tilde{\pi}_i$ remains in Eq. (15) after stating "if $\tilde{\pi}_i = 1/k, \forall i$".
* In Proof 3,  from (16) to (17), it approximates $\nu(z)$ with $\tilde{\nu}(z)$, but is this generally valid? At least, I recommend adding some further discussion.
* I interpreted that setting $\tilde{\pi}_i=1/k, \forall i$ is possible when $\tilde{z}_i$ are independent samples from $\tilde{\nu}$, but how easy is this generally? At least, I recommend adding some further discussion.

## Very minor comments
* On page 12, there is a statement that says "a threshold of θ=5 or θ=6."  Isn't this a mistake, and should it be 0.5 or 0.6?

---

> ### Author Response · Authors · 2025-11-25
> **Response to Reviewer FCkM**
>
> We thank the reviewer for their valuable comments, we address them below:
>
> > My main concern is that without background knowledge on how reconciliation is used, the setup of Section 4. Numerical Tests will be hard to read. Theorems 1-3 abstractly deal with the effects of projecting onto a manifold, and can be understood independently. However, as soon as we get to Section 4.1, it becomes very specific, and readers may be confused by things like generating data using an AR model (Eq. 19) and performing training with LightGBM. Since TMLR readers cover a wide range of machine learning expertise, I think this is a major concern. At least, I was very confused when I first read it. Therefore, I recommend adding a paragraph or subsection in Section 4.1 that clearly explains what kind of data analysis problem is being addressed and how reconciliation is involved, in a way that is easy for those who have not had much experience with reconciliation to understand.
>
> Thanks for this comment, we have now updated the first part of Section 4.1 with a more introductory explanation. The introduced text is the following:
> *The goal of the numerical tests is to evaluate the presented theorems in a realistic forecasting scenario. Reconciliation is usually used as a post-processing technique, after each of the $n$ variables of the multidimensional point $z$ has been predicted independently by a statistical or machine-learning model. Since predictions are independently generated, they won't perfectly lie on the manifold $M$ defined by the constraints. Reconciliation corrects this by projecting an unconstrained forecast onto $M$, enforcing coherence. The experiments in this section simulate a typical forecasting pipeline. We start by generating data from a autoregressive process. This allows us to control the variance and correlation of the different components of $z$ and change them systematically. This is a standard setting in simulation studies for forecast applications. Secondly, we independently fit $n$ standard regression models (LightGBM), which are then used to obtain our unreconciled forecasts $\hat z$. Those are then projected via $\mathscr{S}$, which allows us to investigate the validity of the RMSE reduction theorems. A more technical description of the setup follows.*
> >  Regarding the notation in Theorem 3, it seems somewhat strange that $\tilde{\pi}_i$ remains in Eq. (15) after stating "if $\tilde{\pi}_i=1/k, \forall i$"
>
> We agree, we have now removed the weights and just considered the unweighted Monte Carlo version, as we think is the most common and it eases the notation
>
> > In Proof 3, from (16) to (17), it approximates $\nu(z)$ with $\tilde\nu(z)$, but is this generally valid? At least, I recommend adding some further discussion.
>
> We were using the predictive distribution as best guess for the real distribution, but this can be done only under additional assumption. As pointed out by another reviewer the previous theorem was too weak. We now changed it in a stronger version by adding additional assumptions.
>
> > I interpreted that setting $\tilde{\pi}_i=1/k, \forall i$ is possible when $\tilde{z}_i$ are independent samples from $\tilde{\nu}_i$, but how easy is this generally? At least, I recommend adding some further discussion.
>
> This is usually a standard setting when sampling from a predictive distribution. We have now added the following sentence after the definition of the estimator:  *Without loss of generality, and to ease the notation, we will just consider the unweighted Monte Carlo. *
>
> > On page 12, there is a statement that says "a threshold of $\theta=5$ or $\theta=6$." Isn't this a mistake, and should it be 0.5 or 0.6?
> This is a typo, thank you for catching it

---

> > ### Comment · Reviewer_FCkM · 2025-12-12
> >
> > Thank you for your response.
> >
> > The revision regarding the introduction just before the experiment section is satisfactory to me. Also, the statement around Theorem 3 (in the first draft) is clearer than the previous one.
> >
> > Overall, all of my concerns are satisfactory solved.

---

### Decision · Action_Editor_eedW · 2025-12-19

**Recommendation:** Accept as is

**Additional Comments:**

While the empirical evaluation remains mostly on synthetic data, it is adequate to illustrate the theoretical results within a journal context.

**Audience:**

Yes

**Audience Explanation:**

Forecast reconciliation under nonlinear constraints is a relevant topic for constrained learning. The geometric perspective are likely to interest a subset of TMLR readers, even though this is not a mainstream topic.

**Claims And Evidence:**

Yes

**Claims Explanation:**

The reviewers agree that the theoretical results are correct. The claims are supported by rigorous analysis and, after revision, by clearer exposition. Reviewer concerns were addressed, and the results are sufficient for the scope of TMLR.